# PMIndiaSum: Multilingual and Cross-lingual Headline Summarization for Languages in India

**Ashok Urlana[1,*]**     **Pinzhen Chen[2,*]**     **Zheng Zhao[2]**
**Shay B. Cohen[2]**     **Manish Shrivastava[1]**     **Barry Haddow[2]**

[1]IIIT Hyderabad     [2]University of Edinburgh

ashok.urlana@research.iiit.ac.in, {pinzhen.chen,zheng.zhao}@ed.ac.uk
scohen@inf.ed.ac.uk, m.shrivastava@iiit.ac.in, bhaddow@ed.ac.uk

## Abstract

This paper introduces PMIndiaSum, a multilingual and massively parallel summarization corpus focused on languages in India. Our corpus provides a training and testing ground for four language families, 14 languages, and the largest to date with 196 language pairs. We detail our construction workflow including data acquisition, processing, and quality assurance. Furthermore, we publish benchmarks for monolingual, cross-lingual, and multilingual summarization by fine-tuning, prompting, as well as translate-and-summarize. Experimental results confirm the crucial role of our data in aiding summarization between Indian languages. Our dataset is publicly available and can be freely modified and re-distributed.[1]

## 1 Introduction

The era of deep learning has witnessed great advancements in various natural language processing (NLP) tasks. Yet prevalent solutions usually rely on large datasets, which are limited to high-resource languages. This is particularly pronounced in India, where languages spoken by a large population have been historically overlooked in research and under-resourced (Kumar et al., 2022).

In text summarization, where a system generates a brief description of a longer text, the availability of datasets for Indian languages is restricted in terms of both language coverage and size (Wang et al., 2022).[2] Moreover, some existing datasets are not easily accessible or have been criticized for their quality (Urlana et al., 2022b). Given the multilingual nature of India, having 122 major languages and 22 official ones, the development of

---

[*]Equal contribution.

[1]The corpus is released under the CC BY 4.0 license at https://huggingface.co/datasets/PMIndiaData/PMIndiaSum

[2]This paper refers to the languages widely used in India as "languages in India" or "Indian languages", but they are not exclusive to India and are spoken worldwide.

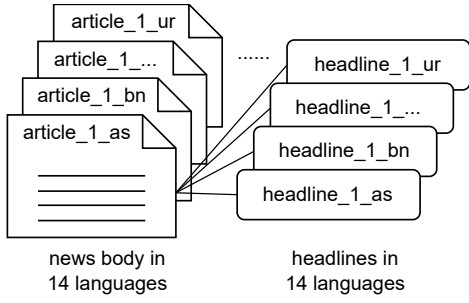

Figure 1: PMIndiaSum supports 14 languages and 196 language pairs, by sourcing from massively parallel articles.

cross-lingual summarization is desirable for information access, however, it is challenging due to the absence of reliable resources.

To address these challenges, we present PMIndiaSum, a massively multilingual and cross-lingual summarization dataset sourced from the Prime Minister of India website.[3] This website publishes political news, usually available in various languages covering the same content. In accordance with established methods (Napoles et al., 2012; Rush et al., 2015), we use a news article as the document and its headline as the summary. Beyond monolingual data, as Figure 1 depicts, the website's multilingualism enables cross-lingual document-summary alignment with a firm confidence in quality.

Our efforts lead to an extensive coverage of 196 language directions from 14 languages across four families, making the corpus the current widest collection of Indian language pairs for summarization. There are 76,680 monolingual document-headline pairs and 620,336 cross-lingual pairs in total. We display language family and code information in Table 1. Of particular note is the inclusion of Manipuri (Meitei), an often neglected language in present-day datasets. The text in our corpus is in Bengali script. Besides the corpus release, we

---

[3]A governmental website that allows content re-distribution. https://www.pmindia.gov.in/en/website-policies

| Family | Language (language code) |
|---|---|
| Dravidian | Kannada (kn), Malayalam (ml), Tamil (ta), Telugu (te) |
| Indo-Aryan | Assamese (as), Bengali (bn), Gujarati (gu), Hindi (hi), Marathi (mr), Odia (or), Punjabi (pa), Urdu (ur) |
| Indo-European | English (en) |
| Tibeto-Burman | Manipuri (mni) |

Table 1: Language family and language code information.

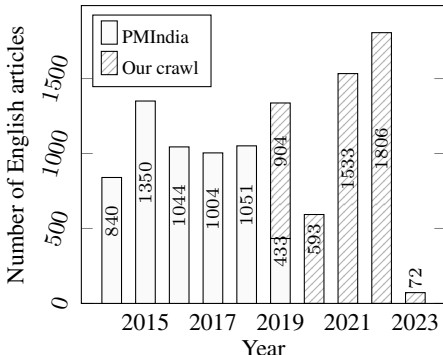

Figure 2: PMIndiaSum acquisition statistics for English, where 56% are from PMIndia and 44% are newly crawled.

obtain results in multiple metrics for popular summarization paradigms like fine-tuning, two-step summarization-translation, and prompting, to serve as a reference for future research.

Though there exist summarization datasets supporting languages in India, some concerns need to be addressed with regard to developing datasets for Indian language summarization. Explicitly, data construction should (1) respect copyright permissions and ensure transparency in data processing; (2) carry out multifaceted quality checks; (3) offer a broad range of language directions; (4) provide reasonably sized data to facilitate system building. Our PMIndiaSum takes into consideration all of these matters. Following Gebru et al. (2021)'s advice, we include a datasheet in Appendix I.

## 2 Preparation of PMIndiaSum

### 2.1 Data acquisition

The Prime Minister of India website posts articles that consist of a headline and a news body. These articles are available in multiple languages, with English being the default option. The HTML structure of each article includes a language indicator and a pointer to the English version. We gather website data for all articles in all available languages, which are sourced from two channels:

1. We ingest readily crawled HTML data from the PMIndia parallel corpus release (Haddow and Kirefu, 2020), which correspond to articles published between 2014 and 2019.
2. We newly crawl for more articles up to early 2023, using PMIndia crawler.[4]

We design a parser to retrieve article bodies and headlines from the HTML. We harvest 94,036 raw article-headline pairs for all languages, which we preliminarily regard as monolingual document-summary data. Figure 2 outlines the origins of English articles across different years.

---
[4]https://github.com/bhaddow/pmindia-crawler

### 2.2 Data processing

To create a high-quality dataset, the collected headline-body pairs undergo rule-based processing in the context of monolingual summarization. The cleaned version contains 76,680 monolingual instances for all languages, which is 81.5% of the raw size. We describe the filtering rules, and list them in Appendix A Table 12 detailing the number of invalid samples removed at each step.

**Language mismatch** Despite confidence in the website's efforts towards language correctness, we set our own language filtering. We discard an entire data instance if either the document or the headline contains text outside of the Unicode range of its designated language.[5] We notice that a large number of samples removed, especially from bn and en, are code-mixed data.

**Duplication and empty** To maintain a dataset with unique document-summary pairs, we remove all duplicates. We also eliminate samples that have identical summaries to steer clear of any headline-article mismatch errors from the website. In addition, we take out instances if either their document or summary is empty.

**Prefix** To enforce that PMIndiaSum is abstractive in nature, we remove all samples where the summary is repeated as the initial or first few sentences in the document. For cases where the summary appears later, the model still needs to "understand" the document in order to "select" a summary. Here, such extraction becomes an easier (but valid) case of summarization.

**Length** We filter out data pairs where the document contains less than two sentences or the sum-

---
[5]Defined under the South and Central Asia-I languages on https://unicode.org/versions/Unicode13.0.0/

| | as | bn | gu | hi | kn | ml | mni | mr | or | pa | ta | te | ur | en |
|---|---|---|---|---|---|---|---|---|---|---|---|---|---|---|
| monolingual size | 2089 | 5557 | 6802 | 7363 | 5307 | 4665 | 4936 | 5816 | 4651 | 4814 | 6079 | 6126 | 4315 | 8160 |
| doc vocab | 73k | 130k | 125k | 94k | 166k | 211k | 143k | 147k | 98k | 80k | 150k | 202k | 61k | 68k |
| sum vocab | 6k | 10k | 14k | 10k | 12k | 12k | 12k | 12k | 9k | 9k | 13k | 16k | 8k | 11k |
| token/doc | 522 | 517 | 505 | 719 | 405 | 336 | 489 | 510 | 544 | 688 | 344 | 430 | 754 | 498 |
| sent/doc | 24.7 | 31.2 | 26.0 | 31.0 | 27.0 | 26.0 | 27.1 | 30.2 | 31.4 | 28.9 | 23.8 | 26.7 | 34.0 | 22.2 |
| token/sum | 11.9 | 11.5 | 12.3 | 15.4 | 12.3 | 10.1 | 13.5 | 11.4 | 10.8 | 18 | 11.7 | 14.1 | 17.7 | 13.4 |
| sent/sum | 1.0 | 1.0 | 1.0 | 1.0 | 1.0 | 1.0 | 1.0 | 1.0 | 1.0 | 1.0 | 1.0 | 1.0 | 1.0 | 1.0 |
| compression | 91.5 | 92.1 | 89.7 | 91.3 | 91.0 | 90.8 | 90.4 | 90.9 | 91.7 | 90.1 | 90.5 | 90.5 | 91.6 | 91.6 |
| density | 6.06 | 3.91 | 5.45 | 8.67 | 4.86 | 3.98 | 7.97 | 4.50 | 3.74 | 8.12 | 5.35 | 4.75 | 8.13 | 7.31 |
| novelty, 1gram | 27.9 | 30.0 | 27.4 | 12.7 | 28.5 | 30.7 | 22.4 | 29.5 | 29.2 | 15.2 | 24.7 | 26.9 | 13.1 | 22.9 |
| novelty, 2gram | 52.5 | 59.8 | 53.5 | 38.7 | 58.4 | 59.2 | 44.9 | 59.0 | 60.8 | 41.3 | 52.9 | 55.8 | 39.7 | 44.9 |
| novelty, 3gram | 63.5 | 73.3 | 65.9 | 54.1 | 70.9 | 72.2 | 59.8 | 71.1 | 73.5 | 57.6 | 66.1 | 71.9 | 56.3 | 56.9 |
| novelty, 4gram | 71.4 | 81.3 | 73.6 | 62.7 | 78.5 | 79.9 | 67.5 | 78.8 | 81.4 | 68.4 | 74.0 | 79.7 | 66.7 | 64.9 |
| redundancy, 1gram | 1.4 | 1.3 | 1.5 | 5.5 | 2.8 | 2.5 | 5.1 | 1.5 | 1.3 | 4.9 | 2.1 | 2.6 | 5.7 | 4.0 |
| redundancy, 2gram | 0.1 | 0.1 | 0.2 | 0.3 | 0.4 | 0.4 | 0.8 | 0.2 | 0.1 | 0.9 | 0.3 | 0.3 | 0.4 | 0.4 |

Table 2: PMIndiaSum monolingual document-summary data analysis.

mary contains less than three tokens, as these pairs may not provide sufficient information for training a summarization model and can become outliers that negatively impact a system's performance. We used the indic_nlp_library for both word and sentence segmentation (Kunchukuttan, 2020).

## 2.3 Monolingual statistics

Table 2 presents the demographics of PMIndiaSum for qualitative inspection. We use token-based measures (Grusky et al., 2018; Narayan et al., 2018). Since many are not well-defined for cross-lingual data, we report the figures for monolingual pairs.

**Vocabulary** We list the count of unique tokens in documents and summaries for each language, under the same tokenization as earlier.

**Length and compression.** Our average document length is 27 sentences with 518 tokens with ur being the longest, and ml being the shortest. Summaries are on average 12 tokens, with nearly all being a single sentence. We then compute the compression ratio, which quantifies how concise a summary $S$ is given its document $D$ as $1 - \frac{\text{len}(S)}{\text{len}(D)}$ in (Bommasani and Cardie, 2020). For each language, we display the average over all samples. A high compression of around 90% implies that the headlines are extremely abstractive and condensed.

**Density** Grusky et al. (2018) introduced extractive fragments $F(D, S)$ as a set of shared sequences in document $D$ and summary $S$. We use their density metric which reflects the degree to which the summary can be composed of texts from the document: $density = \frac{1}{\text{len}(S)} \sum_{f \in F(D,S)} \text{len}(f)^2$.

**Novelty** To evaluate the originality of summaries in our corpus, we compute the percentage of $n$-grams that are present in a summary but not in its corresponding document. We report 1-to-4-gram novelty scores averaged across all samples in each language. Here, unigram novelty is equivalent to $1 - coverage$ in (Grusky et al., 2018).

**Redundancy** Hasan et al. (2021) described redundancy as the amount of repeated information in a summary. It is calculated as the ratio of repetitive n-grams to all $n$-grams in the summary. We measure the average redundancy for unigrams and bigrams, where lower is more desirable.

## 2.4 Multilingualism and parallelism

On top of monolingual data, PMIndiaSum features massive parallelism because the source website publishes a single article in multiple language versions to cater for its audience. This means that document-headline pairs in various languages derived from the same article enable cross-lingual summarization. As depicted in Figure 3, most of the articles are available in at least two languages, and 232 articles are available in all languages. This allows for the creation of cross-lingual and multilingual summarization data pairs.

Technically, every document is paired with summaries in other languages from the same article, and vice versa. Matching is done via the default English pointer in HTML. Such multi-way parallelism results in $14 \times 13 = 182$ cross-lingual pairs in addition to monolingual data. The average data size is 5,477 for monolingual and 3,408 for cross-lingual summarization. Appendix B Table 13 details the sizes for all 196 language directions.

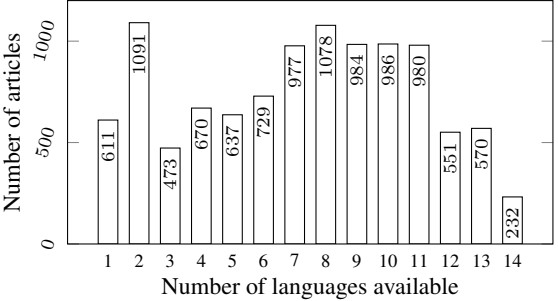

Figure 3: Degree of article parallelism in PMIndiaSum.

|  | en | hi | te |
|---|---|---|---|
| Headlines | **47** | **49** | **49** |
| First sentences | 28 | 45 | 35 |

Table 3: Number of times, out of 50, headlines and first sentences are considered a summary by evaluators.

## 2.5 Data split

To prevent test data leakage in multilingual models, where a model sees an article in one language and is tested on the same article in a different language, we isolate the 232 articles that are available in all languages for validation and testing. We divide the data equally into validation and test sets, resulting in 116 instances for each language pair in each set. This approach provides consistent validation and test splits for each target summary language, regardless of the language directions. All other data are in the training split.

## 2.6 Quality considerations

We have high confidence in text quality since the corpus is crawled from a governmental website and we have specifically designed an HTML parser to remove extraneous elements. In addition, we discuss candidate summaries and data parallelism.

### 2.6.1 Summary choices

We take Rush et al. (2015)'s approach of treating the headline as an article's summary, whereas some works opt for the first sentence instead, e.g. XSum and XL-Sum (Narayan et al., 2018; Hasan et al., 2021). Although the lead sentence can be an overview of an article, this paradigm has received criticism due to potential concerns (Zhao et al., 2020; Urlana et al., 2022b):

1. The first sentence can be part of multi-sentence writing, which is difficult to isolate.
2. Further, the second sentence onwards, when employed as a document, may not always contain all the information in the first sentence.

We conduct an empirical study on the suitability of headlines versus first sentences as summaries. For each of English, Hindi, and Telugu, we invite three native speakers to perform a manual inspection on 50 random samples. Given an article, we ask if the first sentence is a summary and if the

headline is a summary. We record majority votes for each item in Table 3, which suggests that headlines are consistently regarded as a summary; on the contrary, first sentences may not always function as a summary, which is also hard to be automatically identified. In Appendix D, we outline the evaluation protocol and supply a few examples of problematic first sentences. Based on the evaluation outcome, we determine that it is more appropriate to use the headlines as summaries in our corpus. We argue that having the high parallelism between articles, findings from three languages can generalize to the whole corpus.

### 2.6.2 Parallelism

Finally, we assert the validity of cross-lingual alignments. We measure the degree of parallelism by calculating cosine similarity between neural presentations of texts in two languages (LaBSE, Feng et al., 2022). We compute LaBSE scores between summaries as well as between entire documents, from the same article, for each language pair. The average cross-lingual LaBSE score is 0.86 for summaries and 0.88 for documents. These scores indicate the high parallelism between summaries and between documents in different languages; they also notably exceed the 0.74 summary-summary threshold Bhattacharjee et al. (2023) used to extract CrossSum from XL-Sum. Hence, given monolingual document-summary pairs, our choice of substituting the summary (or the document) with one in another language maintains the integrity of cross-lingual pairs. We enclose all pairwise LaBSE scores in Appendix C Table 14 for reference.

## 3 Benchmark Experiments

### 3.1 Task and evaluation

Formally, given a source document $D$, the process of summarization should produce a target summary $S$ with a shorter length, yet conveying the most important message in $D$. We explore three types of models defined by language directions:

1. Monolingual: document $D_L$ and summary $S_L$ are in the same language $L$.

2. Cross-lingual: document $D_L$ and summary $S_{L'}$ are in different languages $L$ and $L'$.

3. Multilingual: monolingual and cross-lingual summarization from $D_{\{L_1, L_2, ..., L_n\}}$ to $S_{\{L_1, L_2, ..., L_n\}}$ within a single model.

In our context, cross-lingual models summarize from one single language to another. Monolingual and cross-lingual models could be more accurate as they concentrate on one language (pair), whereas multilingual models can significantly save storage and computational resources, and likely transfer knowledge across languages.

For evaluation, we report F1 scores of ROUGE-2/L (Lin, 2004), as implemented in XL-Sum, with default language-specific settings such as segmentation and stemming.[6] We also use BLEU-4 from `sacrebleu` (Papineni et al., 2002; Post, 2018).[7] We abbreviate these as R2, RL, and BL hereafter.

## 3.2 Methodology

We intend to provide a benchmark for conventional summarization approaches to cover all language scenarios. We introduce the paradigms below and list the language settings tested with each method in Table 4.

| | Mono | Cross | Multi |
|---|---|---|---|
| Extractive: lead | ✓ | | |
| Extractive: oracle | ✓ | | |
| Summarize-then-translate | | ✓ | |
| Translate-then-summarize | | ✓ | |
| Fine-tuning: full | ✓ | ✓ | ✓ |
| Fine-tuning: zero-shot | | ✓ | |
| LLM prompting | ✓ | | |

Table 4: Summarization approaches and language directions.

**Extractive baseline**  We employ two training-free baselines: 1) selecting the lead sentence, and 2) scoring each sentence in the document against the reference and picking the best in an oracle way.

**Fine-tuning**  Fine-tuning pre-trained language models (PLMs) has shown promising results in summarization for Indian languages (Taunk and Varma, 2022; Urlana et al., 2022a). The paradigm is to load a PLM and further train it for summarization with our data. In monolingual and cross-lingual settings, a PLM is only fine-tuned for a single language direction. On the other hand, in the multilingual setting, we simply mix and shuffle

data for all language pairs, because the data sizes are of the same magnitude across all directions.

**Summarization-and-translation**  In cross-lingual settings, it is practical to leverage a translation system for language conversion. Two common pipelines are 1) summarize-then-translate, where a document is summarized in its original language and then the summary is translated into the target language, and 2) translate-then-summarize, which means first translating the document into the target language followed by summarizing it in that language.

**Zero-shot fine-tuning**  We fine-tune a PLM on all monolingual data only. This serves as an ablation study which only allows same-language document-headline data, but not cross-lingual. We then perform cross-lingual summarization on the test set with this model to measure its zero-shot capability.

**Prompting**  Large language models (LLMs) have demonstrated potential in automatic summarization via prompting (Zhang et al., 2023). On our data, we give a preliminary evaluation of two instruction-finetuned LLMs based on LLaMA-7B (Touvron et al., 2023): Alpaca (Taori et al., 2023) and Vicuna (Chiang et al., 2023). We query an LLM with a prompt "`Article: ${article}. Summarize the article above into a headline in ${language} language. Summary:`", and parse the model's completion as the candidate summary.

**Other techniques**  Lastly, we provide a preliminary discussion on using adapters and summarization models trained on other datasets in Appendix F.

## 3.3 Systems

Our fine-tuning paradigm is tested on two PLMs: IndicBART (Dabre et al., 2022) and mBART-50 (Tang et al., 2021).[8,9] We follow each PLM's convention to add language identification tokens to inform the PLM of the source and target languages. The PLMs support various languages, yet neither supports Manipuri; therefore, we randomly initialize an embedding entry as the `mni` language token in both IndicBART and mBART.[10]

---

[6] https://github.com/csebuetnlp/xl-sum/tree/master/multilingual_rouge_scoring

[7] nrefs:1|case:mixed|eff:no|tok:13a|smooth:exp|version:2.3.1

[8] https://huggingface.co/ai4bharat/IndicBARTSS. We use the IndicBARTSS variant, which deals each language in its own script without the need of mapping to or from Devanagari.

[9] https://huggingface.co/facebook/mbart-large-50

[10] `mni` and `ur` are not covered by IndicBART; `as`, `kn`, `mni`, `or`, and `pa` are not supported by mBART.

| | Lead | | | Oracle | | | IndicBART | | | mBART | | |
|---|---|---|---|---|---|---|---|---|---|---|---|---|
| | R2 | RL | BL | R2 | RL | BL | R2 (Δ) | RL (Δ) | BL (Δ) | R2 (Δ) | RL (Δ) | BL (Δ) |
| as | 31.7 | 41.6 | 18.2 | 34.5 | 44.7 | 23.1 | **41.7** (15.4↑) | **56.1** (18.5↑) | **35.2** (17.4↑) | - | - | - |
| bn | 27.4 | 40.3 | 13.5 | **31.0** | 43.7 | 18.8 | 27.8 (10.5↑) | 46.2 (16.9↑) | 20.5 (10.1↑) | 30.5 (1.3↑) | **48.2** (0.3↑) | 21.5 (1.1↑) |
| gu | 33.7 | 45.3 | 22.7 | 38.4 | 50.9 | 24.6 | 49.8 (4.4↑) | 64.6 (6.7↑) | 44.6 (8.0↑) | **50.1** (1.8↑) | **64.8** (0.3↓) | 45.2 (4.0↑) |
| hi | 44.8 | 58.2 | 30.1 | 49.1 | 62.5 | 33.2 | 53.1 (4.6↑) | 66.9 (6.3↑) | 46.0 (7.3↑) | **55.9** (1.4↑) | **69.4** (0.4↑) | **48.6** (1.3↑) |
| kn | 26.4 | 37.4 | 18.1 | 31.1 | 42.8 | 19.7 | **39.9** (4.3↑) | **56.2** (7.8↑) | **36.3** (7.8↑) | - | - | - |
| ml | 23.8 | 37.1 | 12.7 | **32.7** | 45.8 | **15.4** | 30.3 (6.3↑) | **47.5** (7.5↑) | 15.3 (5.1↑) | 30.2 (3.1↑) | 47.4 (1.8↑) | 14.6 (1.2↓) |
| mni | 38.3 | 50.5 | 26.4 | **42.0** | 54.2 | 26.0 | 38.7 (0.7↓) | 53.0 (1.2↓) | 32.0 (0.8↓) | 41.3 (1.5↑) | **56.4** (0.9↑) | **35.0** (1.6↑) |
| mr | 31.0 | 45.2 | 20.8 | 33.5 | 48.7 | 21.8 | 42.9 (10.5↑) | 59.6 (14.4↑) | 39.7 (14.7↑) | **43.2** (3.0↑) | **61.1** (3.1↑) | **39.8** (4.0↑) |
| or | 21.8 | 37.3 | 11.2 | 26.2 | 42.0 | 15.3 | **38.6** (3.0↑) | **55.4** (6.2↑) | **28.7** (5.1↑) | - | - | - |
| pa | 44.0 | 57.7 | 30.7 | 47.1 | 60.7 | 30.3 | **54.2** (3.4↑) | **68.5** (4.3↑) | **47.5** (4.5↑) | - | - | - |
| ta | 27.9 | 41.6 | 22.9 | 38.0 | 52.8 | 25.5 | **47.5** (6.5↑) | 62.2 (5.8↑) | 37.4 (5.8↑) | 47.4 (5.8↑) | **63.3** (4.1↑) | **37.8** (2.7↑) |
| te | 31.2 | 41.0 | 18.0 | **34.4** | **45.2** | **19.5** | 16.3 (3.0↑) | 32.7 (3.0↑) | 9.9 (1.9↑) | 16.0 (0.4↑) | 33.4 (0.0) | 9.8 (0.8↑) |
| ur | 21.6 | 30.2 | 18.1 | 39.9 | 52.7 | 24.9 | - | - | - | **52.6** (4.3↑) | **64.8** (2.6↑) | **44.0** (3.1↑) |
| en | 33.3 | 42.1 | 18.6 | 38.7 | 48.2 | 22.8 | 63.4 (5.5↑) | 74.5 (5.3↑) | 53.1 (7.7↑) | **66.9** (2.9↑) | **77.8** (1.0↑) | **57.2** (5.6↑) |

Table 5: Monolingual benchmarks: separate models for each language. Bold indicates the best result; (Δ) refers to the change relative to the corresponding multilingual result in Table 9 or Table 10.

| | | R2 | RL | BL |
|---|---|---|---|---|
| fine-tuning | IndicBART | 63.4 | 74.5 | 53.1 |
| | mBART | 66.9 | 77.8 | 57.2 |
| prompting | Alpaca-7B | 20.5 | 35.8 | 14.6 |
| | Vicuna-7B | 30.7 | 48.1 | 16.4 |

Table 6: LLM prompting results for English.

In the summarization-and-translation workflow, monolingual summarization is done using our monolingual fine-tuned PLMs introduced above. The translation process is delegated to a public engine that supports all the involved languages.[11]

Training configurations are detailed in Appendix E. All trainable experiments are conducted three times to obtain the mean and standard deviation statistics. We report average scores, and attach standard deviations in Appendix G.

### 3.4 Experimental results and discussions

#### 3.4.1 Monolingual

We list monolingual results in Table 5. Comparing the two extractive baselines, the oracle performance is better than using the first sentence, yet both are nowhere near perfect scores. This indicates that the summary information is scattered across a document and the headline is abstractive in nature.

Our PLM fine-tuning yields significantly higher numbers than extractions, implying a non-trivial headline summarization task. Generally, mBART is ahead of IndicBART, but it supports fewer languages. In terms of automatic metrics, most languages have reasonable performance except for Telugu. The score lags behind perhaps probably

due to linguistic features, vocabulary, etc, but not the data quality because we also found that Telugu summarization naturally has a lower score than other languages in prior work (Dabre et al., 2022).

The performance of monolingual English summarization from prompting is listed in Table 6 along with fine-tuning baselines. Both Vicuna and Alpaca prompting underperform fine-tuning. Also, we discover that although the LLMs can generate some Indian language texts, the quality is subpar, potentially due to the lack of exposure to these languages during pre-training. We hence suggest that our benchmark cannot be trivially solved by LLM prompting because of language inadequacy and domain specificity. Further research is required to overcome the problems.

#### 3.4.2 Cross-lingual

Fine-tuning all 182 cross-lingual directions is infeasible given our resource constraint. Thus, we shortlist language pairs to fulfil as much as possible: 1) high and low data availability, 2) combinations of language families as in Table 1, and 3) languages supported by both IndicBART and mBART for comparison.

According to results in Table 8, we observe that summarization-and-translation outperforms fine-tuning, but the gaps are not wide. It is worth noting that the comparison is not strictly fair because the summarization-and-translation pipeline uses an external translation model which presumably ingests much more data. Specifically, the order of translation and summarization matters in higher-resource scenarios: for Hindi and English, translation-summarization achieves 10 more

---

[11] https://ssmt.iiit.ac.in/translate

| | en-en | | | | hi-hi | | | | te-te | | | | en-hi | | | | hi-en | | | |
|---|---|---|---|---|---|---|---|---|---|---|---|---|---|---|---|---|---|---|---|---|
| | Mono | | Multi | | Mono | | Multi | | Mono | | Multi | | Cross | | Multi | | Cross | | Multi | |
| | IB | mB | IB | mB | IB | mB | IB | mB | IB | mB | IB | mB | IB | mB | IB | mB | IB | mB | IB | mB |
| **Comprehensibility** | 1 | 0 | 6 | 0 | 0 | 0 | 9 | 0 | 2 | 6 | 5 | 5 | 1 | 1 | **39** | 0 | 4 | 2 | **31** | 1 |
| **Grammar & Fluency** | 0 | 0 | 1 | 0 | 2 | 1 | 0 | 1 | 1 | 1 | 0 | 0 | 0 | 1 | 0 | 1 | 1 | 2 | 1 | 1 |
| **Factuality** | 2 | 1 | 1 | 1 | 4 | 4 | 3 | 3 | 6 | 6 | 6 | 5 | 22 | 26 | 3 | 7 | 21 | 18 | 6 | 10 |
| **Omission** | 18 | 14 | 11 | 15 | **23** | 22 | 12 | 19 | 12 | 18 | 16 | 17 | 11 | 12 | 1 | 15 | **25** | 21 | 7 | 17 |
| **Redundancy** | 5 | 2 | 5 | 2 | **13** | 12 | 11 | 11 | 9 | 7 | 8 | 3 | 9 | 5 | 1 | 3 | 8 | 10 | 1 | 7 |
| **No error** | 28 | **35** | 30 | **34** | 16 | 20 | 17 | 24 | **31** | 25 | 28 | **31** | 13 | 10 | 6 | 20 | 16 | 17 | 11 | 26 |

Table 7: Error analysis on different models and different language scenarios. IB denotes IndicBART and mB denotes to mBART.

points on all three metrics than summarization-translation. The zero-shot columns reveal that both PLMs are unable to perform cross-lingual summarization even though they have been fine-tuned on monolingual data in all languages. This observation indicates the necessity of our true parallel for practical cross-lingual headline summarization between languages in India.

### 3.4.3 Multilingual

We fine-tune both PLMs on all data where the languages are supported. Results are documented in Table 9 for IndicBART and Table 10 for mBART. Regarding cross-lingual cases, multilingual IndicBART seems to only produce sensible numbers for summarization into hi, pa, and en; mBART performs remarkably better than IndicBART.

Referring to the differences in results (Δ) in the monolingual result Table 5, multilingual models are inferior to monolingual models for both IndicBART and mBART. In clear contrast, for 11 out of 12 cross-lingual directions in Table 8, multilingual mBART surpasses separate mBART models fine-tuned for each direction, implying that the availability of data in multiple language pairs helps cross-lingual summarization.

## 4 Analysis and Discussions

### 4.1 Error analysis

To assess and interpret errors made by different models in different language settings, we conducted an error analysis by comparing system outputs with references. We establish error categories and invite human annotators to label the errors in outputs, with setup and explanations of error categories detailed in Appendix H. We randomly sample 50 test outputs from all models that deal with monolingual en, hi, te, as well as en↔hi cross-lingual summarization. Annotation outcome is presented in Table 7.

In monolingual cases, the prevalent error is omission for both IndicBART and mBART models. Conversely, we have different discoveries in cross-lingual models. Dedicated cross-lingual models for a single language pair usually make factuality and omission mistakes. Regarding multilingual systems, IndicBART suffers from language mismatch badly, whereas mBART is associated with factuality, omission and redundancy. Approximately 53% monolingual summaries and 30% cross-lingual summaries are considered to be correct. Although decent automatic scores were seen previously, manual inspection suggests that there is ample room for model improvement on our dataset.

### 4.2 Languages and PLMs

**Resource availability.** In comparison with Indian languages, we see superior summarization scores for English in the monolingual case, probably due to a larger training size. Multilingual models, too, performed better for languages that have more monolingual data on the target side, for instance, en, hi, and gu.

**Unseen language.** Even though Manipuri was not seen during the pre-training of either PLM, it has results close to other languages in monolingual summarization, as well as with multilingual mBART. We hypothesize that this is because the language is written in Bengali script as introduced earlier.

**Language family.** In agreement with previous work (Hasan et al., 2021), both IndicBART and mBART work better for Indo-Aryan languages compared to other families.

**IndicBART versus mBART.** In monolingual and cross-lingual settings, IndicBART is only slightly behind mBART, being only one-third of the size of mBART. However, for multilingual summarization, mBART is clearly preferred in our context.

Table 8: Cross-lingual benchmarks: separate models for each language pair. Bold indicates the best result; (Δ) refers to the change relative to the corresponding multilingual result in Table 9 or Table 10.

| | Summarize-then-translate | | | | | | Translate-then-summarize | | | | | | Fine-tuning | | | | | | Zero-shot | | | | | |
|---|---|---|---|---|---|---|---|---|---|---|---|---|---|---|---|---|---|---|---|---|---|---|---|---|
| | IndicBART | | | mBART | | | IndicBART | | | mBART | | | IndicBART | | | mBART | | | IndicBART | | | mBART | | |
| | R2 | RL | BL | R2 | RL | BL | R2 | RL | BL | R2 | RL | BL | R2 (Δ) | RL (Δ) | BL (Δ) | R2 (Δ) | RL (Δ) | BL (Δ) | R2 | RL | BL | R2 | RL | BL |
| hi–en | 24.9 | 41.6 | 5.8 | 25.5 | 43.1 | 6.3 | 40.5 | **59.8** | 27.0 | 36.3 | 56.8 | 15.8 | **40.6** (24.1↑) | 58.7 (34.7↑) | **28.8** (18.1↑) | 35.3 (8.6↓) | 54.9 (7.2↓) | 24.1 (7.3↓) | 0.0 | 1.0 | 0.1 | 0.3 | 3.5 | 0.2 |
| en–hi | 24.3 | 39.6 | 16.9 | 26.7 | 41.7 | 18.7 | 38.8 | 56.5 | 26.6 | **39.5** | **57.5** | **27.4** | 33.9 (23.1↑) | 51.3 (36.0↑) | 23.2 (16.7↑) | 37.9 (0.2↓) | 55.6 (1.3↓) | 26.8 (0.5↓) | 0.0 | 1.0 | 0.1 | 0.0 | 1.0 | 0.1 |
| gu–te | 6.1 | 19.5 | 1.6 | 6.4 | 19.6 | 1.8 | **16.4** | **32.8** | **4.1** | 9.1 | 24.9 | 2.5 | 14.3 (10.5↑) | 30.4 (20.7↑) | 3.3 (2.8↑) | 14.3 | 30.3 | 2.9 (0.5↓) | 0.1 | 1.1 | 0.1 | 0.1 | 0.9 | 0.1 |
| te–gu | 4.4 | 14.6 | 2.1 | 4.4 | 14.8 | 2.0 | 18.0 | 36.6 | **12.4** | 8.7 | 22.5 | 5.2 | 16.9 (15.4↑) | 37.2 (33.4↑) | 11.6 (11.2↑) | **20.4** (1.2↓) | **40.7** (2.0↓) | 12.4 (1.3↓) | 0.1 | 1.0 | 0.0 | 0.1 | 1.2 | 0.1 |
| ml–mni | **14.9** | **24.7** | **12.5** | 13.5 | 23.7 | 9.6 | 8.2 | 16.2 | 5.5 | 8.8 | 16.4 | 6.7 | 8.2 (2.0↑) | 18.9 (4.2↑) | 4.8 (1.5↑) | 7.6 (8.3↓) | 18.9 (12.5↓) | 2.5 | 0.0 | 0.0 | 0.1 | 0.0 | 0.0 | 0.1 |
| mni–ml | 7.5 | **22.3** | 4.4 | 7.4 | 22.1 | 3.9 | 5.8 | 19.4 | 1.3 | 3.3 | 14.7 | 0.9 | 4.7 (4.7↑) | 14.6 (14.5↑) | 1.9 ( 1.9↑) | 0.6 (10.4↓) | 6.8 (21.1↓) | 0.5 (4.6↓) | 0.0 | 0.1 | 0.0 | 0.0 | 0.0 | 0.1 |
| mr–bn | 13.9 | 32.2 | 8.7 | **14.1** | **33.0** | **9.5** | 12.7 | 31.7 | 7.1 | 11.5 | 31.2 | 7.7 | 12.9 (10.4↑) | 30.4 (23.2↑) | 7.2 (6.3↑) | 12.8 (2.8↓) | 31.5 (3.0↓) | 7.0 (3.2↓) | 0.0 | 0.0 | 0.0 | 0.0 | 0.0 | 0.0 |
| bn–mr | 13.4 | 31.3 | 8.6 | 14.4 | 32.8 | 9.1 | **19.1** | 36.7 | **11.1** | 17.8 | 34.9 | **11.1** | 16.6 (13.4↑) | 35.5 (30.0↑) | 9.7 (8.6↑) | 19.1 (1.3↓) | **37.1** (1.9↓) | 10.5 (1.7↓) | 0.0 | 0.0 | 0.0 | 0.0 | 0.0 | 0.0 |
| te–ta | 14.3 | 30.5 | 9.5 | 13.5 | 31.1 | 9.5 | 24.3 | 41.5 | 19.2 | **25.1** | **43.0** | **20.3** | 17.9 (4.6↑) | 34.8 (5.1↑) | 11.4 (3.4↑) | 16.5 (2.6↓) | 35.7 (4.0↓) | 8.2 (0.5↓) | 0.1 | 1.1 | 0.0 | 0.1 | 1.2 | 0.0 |
| ta–te | 22.5 | 37.8 | 13.4 | 22.3 | **38.4** | **13.6** | 14.0 | 29.6 | 4.8 | 13.2 | 30.1 | 5.3 | 15.7 (7.9↑) | 30.9 (15.0↑) | 2.5 (1.2↑) | 14.2 (0.2↓) | 29.4 (0.4↓) | 2.8 (0.5↓) | 0.1 | 1.4 | 0.1 | 0.1 | 1.3 | 0.1 |
| mni–en | 15.4 | 28.4 | 2.9 | 16.7 | 30.0 | 4.0 | **19.3** | **38.1** | 11.5 | 14.5 | 31.6 | 7.6 | 18.1 (16.4↑) | 35.7 (33.3↑) | **12.3** (11.3↑) | 9.3 (28.8↓) | 22.2 (35.2↓) | 7.2 (19.8↓) | 0.0 | 0.1 | 0.0 | 0.2 | 1.6 | 0.1 |
| en–mni | **24.0** | **35.7** | **18.2** | 23.7 | 35.3 | 18.0 | 12.9 | 22.0 | 9.5 | 14.2 | 23.9 | 11.3 | 7.8 (0.6↓) | 18.4 (1.7↓) | 3.6 (0.1↓) | 5.9 (11.0↓) | 14.9 (18.4↓) | 1.5 (8.6↓) | 0.0 | 0.0 | 0.1 | 0.0 | 0.0 | 0.1 |

Table 9: Multilingual benchmark with IndicBART: a single model for 13 × 13 = 169 supported language pairs.

| | as | | | bn | | | gu | | | hi | | | kn | | | ml | | | mni | | | mr | | | or | | | pa | | | ta | | | te | | | en | | |
|---|---|---|---|---|---|---|---|---|---|---|---|---|---|---|---|---|---|---|---|---|---|---|---|---|---|---|---|---|---|---|---|---|---|---|---|---|---|---|---|
| | R2 | RL | BL | R2 | RL | BL | R2 | RL | BL | R2 | RL | BL | R2 | RL | BL | R2 | RL | BL | R2 | RL | BL | R2 | RL | BL | R2 | RL | BL | R2 | RL | BL | R2 | RL | BL | R2 | RL | BL | R2 | RL | BL |
| as | 26.3 | 37.6 | 17.7 | 3.0 | 9.9 | 1.2 | 7.3 | 10.7 | 3.4 | 9.0 | 12.5 | 5.9 | 3.3 | 6.5 | 2.4 | 2.9 | 8.2 | 0.1 | 10.0 | 23.4 | 4.9 | 2.3 | 4.2 | 1.1 | 5.0 | 7.6 | 1.8 | 10.6 | 16.1 | 6.4 | 0.6 | 1.6 | 0.3 | 4.2 | 9.7 | 0.5 | 12.8 | 18.6 | 7.9 |
| bn | 2.3 | 7.7 | 0.7 | 17.3 | 29.2 | 10.4 | 6.2 | 11.0 | 3.0 | 6.1 | 9.9 | 3.8 | 4.4 | 7.4 | 2.2 | 2.4 | 7.0 | 0.6 | 10.3 | 23.2 | 5.4 | 3.2 | 5.5 | 1.1 | 5.3 | 9.0 | 2.0 | 10.8 | 16.5 | 6.3 | 1.1 | 2.0 | 0.3 | 3.4 | 9.0 | 0.8 | 14.1 | 21.2 | 9.0 |
| gu | 0.4 | 2.5 | 0.2 | 1.3 | 4.3 | 0.3 | 45.4 | 57.9 | 36.6 | 6.4 | 13.6 | 3.2 | 5.0 | 10.4 | 1.2 | 2.5 | 6.3 | 0.3 | 5.1 | 11.9 | 2.7 | 3.3 | 6.1 | 1.1 | 2.1 | 4.4 | 0.7 | 5.1 | 8.1 | 4.2 | 2.4 | 4.7 | 0.7 | 3.8 | 9.7 | 0.5 | 10.0 | 14.6 | 7.8 |
| hi | 1.7 | 5.5 | 0.7 | 0.9 | 4.0 | 0.2 | 5.0 | 9.5 | 1.9 | 60.6 | 80.7 | 48.5 | 5.7 | 10.4 | 3.2 | 6.3 | 15.9 | 0.7 | 7.0 | 16.2 | 3.9 | 6.6 | 13.8 | 2.6 | 5.4 | 9.3 | 1.7 | 17.2 | 20.9 | 11.2 | 4.1 | 7.5 | 1.7 | 6.4 | 14.8 | 0.8 | 14.8 | 24.0 | 10.7 |
| kn | 1.6 | 4.5 | 0.4 | 1.6 | 5.1 | 0.4 | 7.7 | 14.4 | 4.1 | 13.8 | 17.6 | 6.3 | 35.6 | 48.5 | 28.5 | 3.4 | 5.7 | 0.3 | 7.1 | 15.7 | 3.4 | 2.8 | 5.5 | 1.1 | 3.9 | 7.2 | 1.5 | 14.1 | 20.0 | 8.7 | 3.5 | 5.9 | 1.3 | 6.9 | 16.7 | 1.0 | 16.5 | 26.6 | 10.3 |
| ml | 1.5 | 4.7 | 0.5 | 1.3 | 4.8 | 0.4 | 9.3 | 17.0 | 4.3 | 17.6 | 25.0 | 7.0 | 6.4 | 14.0 | 3.7 | 40.0 | 54.0 | 10.2 | 6.2 | 14.7 | 3.3 | 3.8 | 8.2 | 1.5 | 5.7 | 10.7 | 1.9 | 19.1 | 27.9 | 12.9 | 4.7 | 9.6 | 1.4 | 9.7 | 21.3 | 1.9 | 16.8 | 29.8 | 13.4 |
| mni | 0.5 | 3.0 | 0.1 | 0.5 | 4.3 | 0.3 | 0.2 | 0.5 | 0.1 | 0.5 | 0.9 | 0.3 | 0.9 | 1.2 | 0.5 | 0.1 | 0.5 | 0.0 | 39.4 | 54.2 | 32.8 | 0.3 | 0.4 | 0.1 | 1.4 | 2.2 | 0.5 | 0.6 | 0.9 | 0.6 | 0.0 | 0.3 | 0.0 | 0.6 | 1.6 | 0.1 | 1.7 | 2.4 | 1.0 |
| mr | 1.9 | 5.2 | 0.6 | 2.5 | 7.2 | 0.9 | 11.6 | 15.9 | 4.9 | 33.6 | 51.3 | 21.9 | 6.4 | 11.7 | 3.0 | 9.5 | 20.2 | 0.0 | 7.8 | 17.5 | 5.1 | 32.4 | 45.2 | 25.0 | 8.0 | 13.5 | 3.0 | 19.3 | 27.9 | 13.1 | 3.4 | 7.0 | 1.4 | 7.1 | 15.4 | 0.8 | 20.2 | 28.9 | 12.9 |
| or | 2.5 | 5.9 | 0.9 | 2.8 | 7.5 | 0.8 | 8.6 | 15.6 | 4.3 | 17.8 | 28.5 | 8.5 | 4.0 | 8.0 | 2.0 | 4.9 | 11.6 | 0.4 | 8.1 | 18.9 | 5.1 | 5.2 | 8.8 | 2.0 | 35.6 | 49.2 | 23.6 | 15.8 | 22.8 | 11.5 | 3.8 | 6.2 | 1.4 | 6.7 | 13.9 | 0.8 | 14.9 | 22.8 | 8.6 |
| pa | 0.7 | 2.9 | 0.3 | 1.7 | 4.7 | 0.4 | 5.1 | 9.1 | 1.7 | 16.0 | 24.0 | 7.6 | 4.9 | 8.8 | 2.4 | 3.4 | 9.8 | 0.7 | 5.8 | 14.1 | 2.6 | 5.2 | 8.8 | 2.0 | 3.1 | 5.5 | 0.9 | 50.8 | 64.2 | 43.0 | 2.2 | 3.5 | 0.9 | 6.7 | 14.4 | 0.5 | 13.5 | 19.9 | 7.6 |
| ta | 0.8 | 2.6 | 0.3 | 1.7 | 4.9 | 0.4 | 9.5 | 16.8 | 4.9 | 16.7 | 25.1 | 7.1 | 4.7 | 8.2 | 2.0 | 5.1 | 12.5 | 0.7 | 6.4 | 14.8 | 3.2 | 2.7 | 5.3 | 1.0 | 4.4 | 7.2 | 1.5 | 11.7 | 17.0 | 7.4 | 41.0 | 56.4 | 31.6 | 7.8 | 15.9 | 1.3 | 17.1 | 24.5 | 11.7 |
| te | 0.2 | 1.0 | 0.0 | 0.5 | 1.9 | 0.2 | 1.5 | 3.7 | 0.4 | 3.5 | 8.0 | 1.0 | 1.4 | 5.5 | 0.4 | 0.9 | 3.5 | 0.9 | 2.8 | 6.6 | 1.3 | 0.1 | 0.9 | 0.1 | 0.1 | 1.1 | 0.1 | 3.0 | 4.8 | 1.4 | 0.4 | 1.0 | 0.1 | 13.3 | 29.7 | 8.0 | 4.9 | 7.8 | 2.1 |
| en | 0.6 | 2.1 | 0.2 | 1.1 | 5.0 | 0.2 | 7.7 | 15.2 | 3.0 | 15.3 | 24.5 | 6.5 | 5.5 | 10.4 | 2.8 | 4.3 | 11.5 | 0.2 | 7.2 | 16.7 | 3.6 | 3.8 | 8.2 | 1.6 | 3.1 | 6.0 | 0.9 | 12.2 | 18.4 | 7.3 | 4.0 | 6.9 | 1.2 | 6.7 | 14.7 | 0.8 | 57.9 | 69.2 | 45.4 |

Table 10: Multilingual benchmark with mBART: a single model for 10 × 10 = 100 supported language pairs.

| | bn | | | gu | | | hi | | | ml | | | mni | | | mr | | | ta | | | te | | | ur | | | en | | |
|---|---|---|---|---|---|---|---|---|---|---|---|---|---|---|---|---|---|---|---|---|---|---|---|---|---|---|---|---|---|---|
| | R2 | RL | BL | R2 | RL | BL | R2 | RL | BL | R2 | RL | BL | R2 | RL | BL | R2 | RL | BL | R2 | RL | BL | R2 | RL | BL | R2 | RL | BL | R2 | RL | BL |
| bn | 29.2 | 47.9 | 20.4 | 23.5 | 44.4 | 14.6 | 34.3 | 53.9 | 22.9 | 14.1 | 31.7 | 6.9 | 15.4 | 31.7 | 8.5 | 20.4 | 39.0 | 12.2 | 18.7 | 39.1 | 8.7 | 12.6 | 28.5 | 3.5 | 19.3 | 45.6 | 27.9 | 39.8 | 59.1 | 28.9 |
| gu | 16.4 | 35.1 | 9.7 | 48.3 | 65.1 | 41.2 | 36.6 | 55.3 | 25.2 | 14.5 | 31.1 | 6.9 | 15.6 | 31.4 | 8.8 | 21.3 | 39.8 | 13.5 | 19.1 | 39.1 | 9.1 | 13.1 | 29.5 | 3.4 | 20.0 | 45.8 | 28.2 | 42.9 | 61.0 | 30.3 |
| hi | 15.9 | 33.9 | 9.7 | 24.8 | 46.0 | 15.6 | 54.5 | 69.0 | 47.3 | 14.0 | 31.4 | 6.4 | 15.7 | 31.8 | 9.3 | 22.4 | 40.8 | 14.9 | 19.8 | 40.4 | 9.8 | 15.0 | 31.7 | 3.7 | 20.4 | 46.5 | 28.7 | 43.9 | 62.1 | 31.4 |
| ml | 15.4 | 34.7 | 9.3 | 22.7 | 43.4 | 13.9 | 36.2 | 54.8 | 24.6 | 27.1 | 45.6 | 15.8 | 15.9 | 31.4 | 8.9 | 19.6 | 38.4 | 12.3 | 18.6 | 38.5 | 9.0 | 12.3 | 29.6 | 3.6 | 19.0 | 45.2 | 27.3 | 41.3 | 60.5 | 29.1 |
| mni | 12.0 | 30.3 | 7.1 | 21.2 | 41.3 | 13.6 | 32.3 | 51.3 | 21.9 | 11.0 | 27.9 | 5.1 | 39.8 | 55.5 | 33.4 | 17.8 | 36.5 | 11.0 | 17.1 | 36.1 | 8.6 | 13.7 | 27.9 | 2.8 | 18.2 | 43.1 | 25.9 | 38.1 | 57.4 | 27.0 |
| mr | 15.6 | 34.5 | 10.2 | 24.5 | 45.7 | 15.1 | 38.2 | 57.0 | 26.3 | 14.4 | 31.2 | 6.4 | 15.6 | 31.4 | 8.4 | 40.2 | 58.0 | 35.8 | 18.8 | 39.5 | 10.7 | 14.4 | 30.0 | 3.9 | 18.5 | 44.5 | 26.7 | 41.9 | 61.0 | 30.3 |
| ta | 15.2 | 33.0 | 8.4 | 21.2 | 41.9 | 12.6 | 34.0 | 52.9 | 23.1 | 13.1 | 29.8 | 6.0 | 13.4 | 29.1 | 6.2 | 19.6 | 37.8 | 12.5 | 41.6 | 59.2 | 35.1 | 14.4 | 29.8 | 3.2 | 18.0 | 44.2 | 26.4 | 40.7 | 59.7 | 29.1 |
| te | 15.1 | 32.9 | 9.4 | 21.6 | 42.7 | 13.7 | 36.1 | 55.1 | 23.5 | 13.1 | 30.2 | 6.6 | 16.2 | 31.7 | 9.3 | 21.1 | 39.5 | 13.7 | 19.1 | 39.7 | 11.6 | 29.8 | 33.4 | 9.0 | 18.4 | 44.7 | 26.7 | 40.5 | 59.7 | 29.5 |
| ur | 14.5 | 32.8 | 8.8 | 23.8 | 44.6 | 15.3 | 34.7 | 54.0 | 23.0 | 12.7 | 29.5 | 6.2 | 14.5 | 30.6 | 7.7 | 20.3 | 39.0 | 12.1 | 19.1 | 39.6 | 11.2 | 14.3 | 29.9 | 3.2 | 62.2 | 48.3 | 48.3 | 41.9 | 60.3 | 30.6 |
| en | 15.7 | 34.3 | 9.5 | 25.1 | 46.4 | 15.5 | 38.1 | 56.9 | 26.3 | 13.6 | 31.1 | 6.0 | 16.9 | 33.3 | 10.1 | 20.7 | 39.4 | 13.4 | 20.2 | 40.6 | 11.9 | 14.3 | 30.5 | 4.0 | 21.3 | 48.0 | 29.9 | 64.0 | 76.7 | 51.5 |

## 5 Related Work

Document-summary pairs are the key components of a summarization dataset. A typical acquisition workflow is to utilize publicly available articles, given their high availability. To form a document-summary pair, an article can be paired with either human-written summaries, or accompanying texts such as the headline or the first sentence.

### 5.1 Data acquisition

Earlier works like the CNN/DM corpus utilized English news articles as documents and highlights as summaries (Hermann et al., 2015; Nallapati et al., 2016), followed by Scialom et al. (2020) to construct MLSUM in five Indo-European languages. Hasan et al. (2021) created XL-Sum with BBC articles' first sentence as a summary and the rest as a document. Varab and Schluter (2021) developed MassiveSumm by scraping a wide range of websites. The ILSUM dataset crawled several leading newspapers in India (Satapara et al., 2022), and TeSum used Telugu news with summaries from human annotators (Urlana et al., 2022b). Verma et al. (2023) built a multilingual multi-modal dataset M3LS from the BBC website. Recently, a concurrent work sourced data from a news aggregator to form a multilingual headline dataset in Indic languages (Vārta, Aralikatte et al., 2023).

Exploiting public resources usually results in same-language document-summary pairs as introduced above. Zhu et al. (2019) translated summaries in monolingual datasets into another language while ensuring that the round-trip translation of the summary did not deviate from the original one. The CrossSum corpus (Bhattacharjee et al., 2023) was derived from XL-Sum by thresholding similarity between cross-lingual summaries, and pairing a summary with the document of a parallel summary. The naturally arising cross-lingual datasets include WikiLingua which aligned summaries and documents using image pivots (Ladhak et al., 2020) and EUR-Lex-Sum from a platform hosting legal documents (Aumiller et al., 2022).

Our PMIndiaSum falls in the category of using public article-headline pairs. The data itself is massively cross-lingual due to the multilingualism of the published news. We greatly benefit from the materials released by the PMIndia parallel corpus (Haddow and Kirefu, 2020), which is a machine translation dataset between English and Indian languages, extracted from the same website.

### 5.2 Coverage for languages in India

The current availability of Indian language summariztaion data is limited. TeSum is monolingual, ILSUM covers three Indian languages, XL-Sum supports eight, and M3LS covers ten. Covering the largest number of Indian languages are MassiveSumm and Vārta, but these works release URLs and processing scripts instead of the data; copyright holders' conditions and licenses remain unknown. Such practices transfer the risk and responsibility to dataset users. Our data can be freely used, modified, and re-distributed.

Moreover, the above multilingual datasets cannot be used for cross-lingual research. Existing data for summarization between Indian languages are scarce. WikiLingua supports one Indian language, and to the best of our knowledge, before ours, CrossSum is the largest which contains 56 language pairs from pairing up 8 languages in XL-Sum. In comparison, even excluding English, our PMIndiaSum contains 13 Indian languages and 156 cross-lingual pairs. In a zero-shot fashion, one can train on multilingual datasets for cross-lingual capability. Nonetheless, the performance may be inferior as we have shown. More importantly, monolingual and multilingual datasets cannot provide a benchmark to evaluate cross-lingual research. Our work fills the gap by providing both training data and a trustworthy test suite for a large number of Indian language directions.

## 6 Conclusion and Future Work

We have presented PMIndiaSum, a headline summarization corpus focused on 14 languages in India, supporting monolingual, cross-lingual, and multilingual summarization in 196 directions. We outlined the processing steps, explained data statistics, discussed quality considerations, and made the dataset publicly accessible. We also published benchmarks for extractive baselines, summarize-and-translate, and fine-tuning, involving two pre-trained language models. Experiment results emphasize the value of our dataset for summarization between languages in India.

A natural extension is to continuously integrate new articles from the source website to update the dataset. However, to ensure that experimental comparisons remain fair, one needs to maintain consistent data sizes in model training. Another direction is to ingest more websites to expand the size, domain coverage, language availability, and modality.

## Limitations

Our data sources are scraped from a government website, leading to a domain bias towards political news, and a style bias towards headline-like summaries. Also, we selected the articles available in all languages to be validation and testing instances, to prevent information leakage. While we believe this to be a sensible decision, those articles might carry a distributional drift, such as being more important for all readers or easier to translate.

## Ethics Statement

We place trust in the Indian government website to eliminate inappropriate content, but we could not inspect each data instance ourselves. On the other hand, we note that the values promoted by a governmental agency might not align with every potential user of the data. Also, the provision of data in widely spoken languages might further edge out lesser-used languages in research.

## Acknowledgements

We thank Dennis Aumiller and the anonymous reviewers for giving feedback. We are grateful to Pruthwik Mishra, Lokesh Madasu, Kalahasti Ganesh Srivatsa, and Nikhilesh Bhatnagar for helping with human evaluation. We also thank Marcio Fonseca for providing the code for LLM prompting, and LTRC, IIIT Hyderabad for providing the translation engine for our summarization-and-translation experiments. Our work would not be possible if the Prime Minister's Office of the Government of India or the authors of the PMIndia parallel corpus did not allow the re-distribution of their materials.

Pinzhen Chen and Barry Haddow are funded by UK Research and Innovation (UKRI) under the UK government's Horizon Europe funding guarantee [grant numbers 10052546 and 10039436]. Zheng Zhao is supported by the UKRI Centre for Doctoral Training in Natural Language Processing (EP/S022481/1).

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

## A    Data filtering details

We manually inspected the data and discovered that articles labeled as English may contain non-English content, despite having English URLs. We used these URLs as references to match cross-lingual pairs, but we didn't assume the content was always in English. Therefore, we cleaned our data strictly by language. Table 12 shows the number of invalid samples removed at each step, together with the raw size and the cleaned size.

## B    Data size breakdown

Table 13 lists the total size for each language pair. Rows correspond to the document language and columns correspond to the summary language. Val-idation and test sizes are 116 each, and the training split employs the rest.

## C    LaBSE similarity breakdown

Technically, we obtain LaBSE representations by feeding a summary or document into the model, truncated at the maximum length if necessary. Ta-ble 14 presents the LaBSE scores between cross-lingual documents and between cross-lingual sum-maries. Rows and columns indicate the language pair. The upper right triangle contains scores for summary pairs, and the lower left triangle contains scores for document pairs. Due to the lack of sup-port for Manipuri in multilingual BERT (Devlin et al., 2019), scores involving Manipuri are consid-erably lower than other language combinations.

## D    Human evaluation details and samples

We carry out a human evaluation to compare head-lines and first sentences as summaries. For each of English, Hindi, and Telugu, we ask three native speakers to consider the accuracy, informativeness, and quality of the potential summaries from 50 samples based on the following guideline.

|  | en | hi | te |
|---|---|---|---|
| Headlines | 0.80 | 0.52 | 0.54 |
| First sentences | 0.82 | 0.88 | 0.67 |

Table 11: Inter rater reliability (ICC).

In this task, you will look at a set of articles and summaries. Each article will be presented with two (2) potential summaries. The aim of the evaluation is to identify whether the presented text is a summary. Below are the steps you need to take.

1. Read the article to understand the context.
2. Read both summaries carefully. Consider the accuracy, informativeness, and quality of the summaries. We provide the definition of these criteria below.
   - *Accuracy*: how closely the summary reflects the factual content of a news article. An accurate summary should not include any false information or misrepresentations of the content.
   - *Informativeness*: how much information the summary provides about the main points of the news article. An informative summary should provide enough detail to give the reader a good understanding of the article's content.
   - *Quality*: the overall standard of the summary. A high-quality summary should be fluent, well-written, free of errors, and easy to read.
3. Provide your binary decision on whether each text is a summary. Base your decision solely on the quality, accuracy, and informativeness of the content, without being influenced by factors such as writing style, personal preferences, etc. Please note that an incomplete sentence is acceptable as long as it does not compromise accuracy, informativeness, or quality.

For each sample document, we record the majority voting (yes/no) from three annotators in Section 2.6.1 Table 3. Alongside the results, in Table 11, we compute the inter-rater reliability using Intra-class Correlation Coefficient as per (ICC, Koo and Li, 2016; Shrout and Fleiss, 1979). We observe that the annotators are in moderate to good agreement for all languages and both potential summaries.

We also note samples from the dataset to show potential problems with using first sentences as summaries in Table 17. Sometimes the first sentence is part of a speech expressing gratitude.

## E    Experimental setup

For prompting, our initial finding is that open-source LLMs do not work very well for non-English, so our prompting experiments are for English only. In terms of technical details, IndicBART and mBART-large-50 models have 244M and 610M parameters each. For fine-tuning, we set a training budget of 100 epochs and apply early stopping after three consecutive non-improving cross-entropy during validation. We use an effective batch size of 96 by combining different batch sizes, numbers of GPUs, and gradient accumulation. For document-summary pairs, we set the maximum lengths to 1024 and 64 respectively. All other configurations follow the Hugging Face trainer default.[12] We do not perform hyperparameter search. Specifically, our GPUs include Nvidia GeForce RTX 2080 Ti (11GB), GeForce RTX 3090 (24GB), and A100 (80GB). Models mostly converge within 10 epochs.

## F    Other summarization techniques

### F.1    Adapters

Adapters are small trainable modules inserted into a giant PLM; during PLM fine-tuning, the entire model can be frozen except for the adapters (Houlsby et al., 2019). This paradigm achieves great parameter efficiency as only the adapter weights need to be updated and saved.

Adapters are relevant to summarization research, but we omit experiments on these because Zhao and Chen (2022) show that with a few thousand training data, the performance of adapters is not comparable to fine-tuning a PLM. We, nevertheless, encourage future research to try this out.

### F.2    Models trained on another dataset

An effortless approach apart from extracting the first sentence from a document is to use a summarization model that has been trained on another dataset. In this regard, we experiment with an mT5-based multilingual summarization model trained on the XL-Sum dataset comprising 44 languages.[13] We tested it on our PMIndiaSum monolingual test sets. As Table 15 shows, the performance of the XL-Sum pre-trained model is subpar on our data, potentially due to domain mismatch. This implies that our test sets cannot be easily solved by training on other datasets.

---

[12]https://huggingface.co/docs/transformers/main_classes/trainer
[13]https://huggingface.co/csebuetnlp/mT5_multilingual/_XLSum

| | as | bn | gu | hi | kn | ml | mni | mr | or | pa | ta | te | ur | en |
|---|---|---|---|---|---|---|---|---|---|---|---|---|---|---|
| raw size | 2925 | 9250 | 7393 | 7642 | 5743 | 5106 | 5432 | 6034 | 4760 | 7315 | 6314 | 6656 | 5622 | 13844 |
| - language mismatch | 68 | 3418 | 302 | 80 | 361 | 344 | 271 | 93 | 40 | 2367 | 68 | 422 | 1144 | 5288 |
| - duplicate pairs | 4 | 11 | 11 | 5 | 16 | 7 | 1 | 12 | 8 | 11 | 23 | 8 | 6 | 2 |
| - duplicate summaries | 8 | 28 | 34 | 110 | 38 | 25 | 14 | 54 | 26 | 53 | 54 | 34 | 24 | 202 |
| - empty | 0 | 1 | 0 | 0 | 0 | 0 | 2 | 0 | 1 | 0 | 0 | 0 | 0 | 0 |
| - prefixes | 33 | 37 | 28 | 49 | 13 | 52 | 92 | 28 | 13 | 30 | 73 | 46 | 133 | 169 |
| - doc <2 sentences | 723 | 191 | 216 | 35 | 5 | 12 | 113 | 30 | 21 | 40 | 13 | 19 | 0 | 19 |
| - sum <3 tokens | 0 | 7 | 0 | 0 | 3 | 1 | 3 | 1 | 0 | 0 | 4 | 1 | 0 | 4 |
| **final (monolingual)** | 2089 | 5557 | 6802 | 7363 | 5307 | 4665 | 4936 | 5816 | 4651 | 4814 | 6079 | 6126 | 4315 | 8160 |

Table 12: Data preprocessing and filtering statistics for document-summary pairs in each language.

| $\frac{S_{Lj}}{D_{Li}}$ | as | bn | gu | hi | kn | ml | mni | mr | or | pa | ta | te | ur | en |
|---|---|---|---|---|---|---|---|---|---|---|---|---|---|---|
| as | 2089 | 1153 | 1728 | 1716 | 1411 | 1166 | 1737 | 1415 | 1711 | 1332 | 1607 | 1570 | 1571 | 1743 |
| bn | 1153 | 5557 | 4655 | 3987 | 3573 | 3332 | 2898 | 4268 | 3079 | 3479 | 3845 | 4239 | 2345 | 4075 |
| gu | 1728 | 4655 | 6802 | 4896 | 4411 | 3901 | 3998 | 4940 | 4021 | 4052 | 4731 | 5111 | 3185 | 5216 |
| hi | 1716 | 3987 | 4896 | 7363 | 3680 | 3723 | 3163 | 4340 | 3784 | 3426 | 4817 | 4376 | 3061 | 5899 |
| kn | 1411 | 3573 | 4411 | 3680 | 5307 | 3176 | 3405 | 3800 | 3439 | 3526 | 3720 | 4151 | 2742 | 4325 |
| ml | 1166 | 3332 | 3901 | 3723 | 3176 | 4665 | 2423 | 3547 | 3037 | 2881 | 3957 | 3737 | 2299 | 3769 |
| mni | 1737 | 2898 | 3998 | 3163 | 3405 | 2423 | 4936 | 3120 | 2901 | 3198 | 3181 | 3596 | 3199 | 3807 |
| mr | 1415 | 4268 | 4940 | 4340 | 3800 | 3547 | 3120 | 5816 | 3450 | 3573 | 4219 | 4488 | 2582 | 4411 |
| or | 1711 | 3079 | 4021 | 3784 | 3439 | 3037 | 2901 | 3450 | 4651 | 3224 | 3722 | 3667 | 2646 | 3873 |
| pa | 1332 | 3479 | 4052 | 3426 | 3526 | 2881 | 3198 | 3573 | 3224 | 4814 | 3356 | 3772 | 2489 | 3615 |
| ta | 1607 | 3845 | 4731 | 4817 | 3720 | 3957 | 3181 | 4219 | 3722 | 3356 | 6079 | 4514 | 3139 | 4909 |
| te | 1570 | 4239 | 5111 | 4376 | 4151 | 3737 | 3596 | 4488 | 3667 | 3772 | 4514 | 6126 | 2847 | 4955 |
| ur | 1571 | 2345 | 3185 | 3061 | 2742 | 2299 | 3199 | 2582 | 2646 | 2489 | 3139 | 2847 | 4315 | 3415 |
| en | 1743 | 4075 | 5216 | 5899 | 4325 | 3769 | 3807 | 4411 | 3873 | 3615 | 4909 | 4955 | 3415 | 8160 |

Table 13: PMIndiaSum data sizes for all language pairs. Each cell corresponds to the size of a dataset $(D_{Li}, S_{Lj})$, with source document $D$ in language $Li$ and target summary $S$ in language $Lj$.

## G Standard deviations

Standard deviations are listed in Table 16 for monolingual summarization, Table 19 for cross-lingual summarization, Table 20 for multilingual IndicBART, and Table 21 for multilingual mBART. These correspond to the mean ROUGE and BLEU scores reported in Section 3.4. We do not notice any peculiar numbers.

## H Error analysis details

We conduct an error analysis on the outputs of monolingual and multilingual models for English, Hindi, and Telugu. Additionally, we analyzed the outputs of cross-lingual and multilingual models for English-to-Hindi and Hindi-to-English summarization. We define our error categories in Table 18 which displays each error type with an explanation and an example. As per our definition, a single summary output may fall into multiple error categories.

It is important to highlight the distinction between this error analysis and summary quality analysis in previous works (Grusky et al., 2018; Clark et al., 2023). In our analysis, human annotators identify errors in the system outputs by referring to the gold output rather than the source document. To ensure accuracy, our annotators are native speakers of the language of the summaries. In the case of cross-lingual summaries, they were also bilingual in the source language.

| $L_i$ \ $L_j$ | as | bn | gu | hi | kn | ml | mni | mr | or | pa | ta | te | ur | en |
|---|---|---|---|---|---|---|---|---|---|---|---|---|---|---|
| as | - | 0.88 | 0.86 | 0.82 | 0.87 | 0.87 | 0.80 | 0.85 | 0.90 | 0.87 | 0.85 | 0.88 | 0.85 | 0.87 |
| bn | 0.89 | - | 0.86 | 0.85 | 0.87 | 0.86 | 0.73 | 0.87 | 0.88 | 0.87 | 0.85 | 0.86 | 0.87 | 0.86 |
| gu | 0.85 | 0.90 | - | 0.92 | 0.91 | 0.88 | 0.70 | 0.90 | 0.90 | 0.92 | 0.88 | 0.90 | 0. 91 | 0.89 |
| hi | 0.81 | 0.89 | 0.93 | - | 0.89 | 0.86 | 0.65 | 0.91 | 0.87 | 0.91 | 0.88 | 0.88 | 0.90 | 0.89 |
| kn | 0.86 | 0.91 | 0.94 | 0.92 | - | 0.89 | 0.72 | 0.90 | 0.89 | 0.91 | 0.89 | 0.91 | 0.89 | 0.90 |
| ml | 0.88 | 0.90 | 0.91 | 0.89 | 0.93 | - | 0.73 | 0.88 | 0.89 | 0.89 | 0.87 | 0.88 | 0.88 | 0.88 |
| mni | 0.84 | 0.75 | 0.71 | 0.68 | 0.73 | 0.75 | - | 0.69 | 0.74 | 0.71 | 0.71 | 0.74 | 0.69 | 0.74 |
| mr | 0.85 | 0.90 | 0.93 | 0.93 | 0.92 | 0.90 | 0.71 | - | 0.89 | 0.90 | 0.88 | 0.89 | 0.89 | 0.89 |
| or | 0.90 | 0.91 | 0.92 | 0.90 | 0.92 | 0.92 | 0.76 | 0.91 | - | 0.90 | 0.88 | 0.89 | 0.88 | 0.90 |
| pa | 0.87 | 0.90 | 0.94 | 0.93 | 0.93 | 0.91 | 0.73 | 0.92 | 0.92 | - | 0.89 | 0.90 | 0.91 | 0.90 |
| ta | 0.85 | 0.90 | 0.91 | 0.90 | 0.93 | 0.91 | 0.73 | 0.91 | 0.91 | 0.91 | - | 0.89 | 0.88 | 0.88 |
| te | 0.87 | 0.91 | 0.92 | 0.90 | 0.94 | 0.92 | 0.75 | 0.91 | 0.92 | 0.93 | 0.92 | - | 0.89 | 0.90 |
| ur | 0.84 | 0.89 | 0.92 | 0.90 | 0.91 | 0.90 | 0.72 | 0.90 | 0.90 | 0.92 | 0.90 | 0.91 | - | 0.90 |
| en | 0.84 | 0.89 | 0.90 | 0.91 | 0.91 | 0.90 | 0.72 | 0.90 | 0.90 | 0.90 | 0.90 | 0.90 | 0.89 | - |

Table 14: LaBSE scores between parallel documents ($D_{Li}$, $D_{Lj}$, bottom left) and parallel summaries ($S_{Li}$, $S_{Lj}$, upper right).

|  | R2 | RL | BL |
|---|---|---|---|
| bn | 16.9 | 30.0 | 9.8 |
| gu | 21.2 | 33.3 | 15.0 |
| hi | 23.4 | 40.1 | 16.4 |
| mr | 21.1 | 36.2 | 14.8 |
| pa | 29.7 | 45.1 | 23.2 |
| ta | 19.7 | 35.6 | 12.8 |
| te | 7.7 | 17.9 | 4.1 |
| ur | 26.3 | 40.4 | 20.1 |
| en | 11.2 | 24.0 | 5.5 |

Table 15: Results of a multilingual model trained on XL-Sum and tested on our PMIndiaSum.

|  | IndicBART | | | mBART | | |
|---|---|---|---|---|---|---|
|  | R2 | RL | BL | R2 | RL | BL |
| as | 0.6 | 0.5 | 0.6 | - | - | - |
| bn | 1.0 | 1.0 | 1.0 | 0.2 | 0.3 | 0.7 |
| gu | 1.5 | 1.4 | 0.1 | 1.3 | 1.5 | 1.6 |
| hi | 0.8 | 0.8 | 0.6 | 0.5 | 0.4 | 1.8 |
| kn | 0.6 | 0.8 | 0.2 | - | - | - |
| ml | 0.8 | 0.6 | 0.3 | 1.6 | 1.4 | 1.1 |
| mni | 0.3 | 0.7 | 0.5 | 1.9 | 2.7 | 1.7 |
| mr | 1.4 | 1.4 | 2.7 | 1.3 | 1.1 | 1.2 |
| or | 1.5 | 1.2 | 1.5 | - | - | - |
| pa | 0.7 | 0.6 | 0.7 | - | - | - |
| ta | 0.9 | 0.5 | 0.2 | 0.7 | 0.4 | 2.1 |
| te | 0.4 | 0.9 | 0.0 | 1.3 | 1.1 | 0.7 |
| ur | - | - | - | 2.8 | 2.1 | 2.7 |
| en | 0.4 | 0.00 | 1.8 | 2.0 | 1.1 | 2.1 |

Table 16: Monolingual benchmark: standard deviations.

**Example 1**

**en document:** Your Excellencies President Xi Jinping, President Jacob Zuma, President Michel Temer, President Vladimir Putin, Let me begin by sincerely thanking President Xi again for his warm reception and the excellent organisation of this Summit. Our interaction during the restricted session was constructive. It enriched our mutual understanding and perspectives. After more than a decade of existence, BRICS has developed a robust framework for cooperation. ...

**en headline:** PM's Intervention at the Plenary Session of 9th BRICS Summit, Xiamen, China

**hi document:** महामहिम राष्ट्रपति शी जिनपिंग राष्ट्रपति जैकब जूमा राष्ट्रपति माइकल टेमर राष्ट्रपति ब्लादिमीर पुतिन मैं राष्ट्रपति शी का जोरदार मेजबानी और इस शिखर सम्मेलन के शानदार आयोजन के लिए शुक्रिया अदा कहना चाहता हूं। हमारी पिछले सत्रों में बातचीत सार्थक रही थी, जिसने हमारी परस्पर समझ और दृष्टिकोण को मजबूत किया है। अपने अस्तित्व में आने के एक दशक बाद ब्रिक्स ने सहयोग के लिए एक मजबूत ढांचा विकसित किया है। ...

**hi headline:** चीन के शियामेन में आयोजित नौवें ब्रिक्स शिखर सम्मेलन के पूर्ण अधिवेशन में प्रधानमंत्री का संबोधन

**te document:** శ్రేష్ఠులైన అధ్యక్షులు శ్రీ శీ జిన్పింగ్, అధ్యక్షులు శ్రీ జాకబ్ జుమా, అధ్యక్షులు శ్రీ మైఖేల్ టెమెర్, అధ్యక్షులు శ్రీ వ్లాదిమీర్ పుతిన, ఈ శిఖర సమ్మేళనానికి సాదరంగా ఆహ్వానించినందుకు మరియు ఈ సమ్మేళనాన్ని ఉత్తమమైన రీతిలో నిర్వహిస్తున్నందుకుగాను అధ్యక్షుల వారు శ్రీ శీ కి తొలుత ధన్యవాదాలు తెలియజేస్తూ నా ఈ ప్రసంగాన్ని ప్రారంభిస్తున్నాను. మా మధ్య పరిమిత స్థాయి సమావేశం సందర్భంగా జరిగినప్పటి సంభాషణ ఫలప్రదంగా ముగిసింది. అది మా ఇరువురి దృష్టి కోణాలను, పరస్పర అవగాహనను సుసంపన్నం చేసింది. ...

**te headline:** చైనా లోని జియామెన్ లో 2017 సెప్టెంబర్ 4వ తేదిన 9వ బ్రిక్స్ శిఖర సమ్మేళనం యొక్క సర్వ సభ్య సదస్సు లో ప్రధాన మంత్రి శ్రీ నరేంద్ర మోదీ ప్రసంగ పాఠం

---

**Example 2**

**en document:** Bharat Mata Ki Jai! Bharat Mata Ki Jai! The Governor of Tripura Shri Satyadev Arya ji, Tripura's young and energetic Chief Minister Shri Biplab Deb ji, Tripura's Deputy Chief Minister Shri Jishnu Dev Varma ji, my Cabinet colleagues Sister Pratima Bhoumik ji and Shri Jyotiraditya Scindia ji, Ministers in the State Government Shri NC Debbarma ji, Shri Ratanlal Nath ji, Shri Pranjit Singha Roy ji and Shri Manoj Kanti Deb ji, other public representatives and my dear brothers and sisters who have come in large numbers! Greetings to you all. My best wishes to you all on the new year 2022! In the beginning of the year itself, Tripura is receiving three gifts today with the blessings of Maa Tripura Sundari. ...

**en headline:** PM's speech at the inauguration of Maharaja Bir Bikram Airport and other projects in Tripura

**hi document:** भारत माता की जय! भारत माता की जय! त्रिपुरा के राज्यपाल श्री सत्यदेव आर्य जी, यहां के युवा और ऊर्जावान मुख्यमंत्री श्री बिप्लब देब जी, त्रिपुरा के उप-मुख्यमंत्री श्री जिष्णु देव वर्मा जी, केंद्रीय मंत्रीमंडल में मेरे सहयोगी बहन प्रतिमा भौमिक जी, श्री ज्योतिरादित्या सिंधिया जी, राज्य सरकार में मंत्री श्री एनसी देबबर्मा जी, श्री रतनलाल नाथ जी, श्री प्रणजीत सिंघा रॉय जी, श्री मनोज कांति देब जी, अन्य जनप्रतिनिधिगण और विशाल संख्या में पधारे हुए मेरे प्यारे भाइयों और बहनों ! शबाई के नमोस्कार। शकल के दू हजार बाइस वर्षेर अनेक-अनेक शुभेच्छा। जोतोनो खूनुमखा। जोतोनो बीशी कांतालनी खा काहाम याफर ओ। साल की शुरुआत में ही, त्रिपुरा को मां त्रिपुर सुंदरी के आशीर्वाद से आज तीन उपहार मिल रहे हैं। ...

**hi headline:** त्रिपुरा में महाराजा बीर बिक्रम हवाई अड्डे और अन्य परियोजनाओं के उद्घाटन के अवसर पर प्रधानमंत्री का भाषण

**te document:** భారత్ మాతా కీ జై ! భారత్ మాతా కీ జై ! త్రిపుర గవర్నర్ శ్రీ సత్యదేవ్ ఆర్య గారు, త్రిపుర యువ శక్తివంతమైన ముఖ్యమంత్రి శ్రీ బిప్లబ్ దేబ్ జీ, త్రిపుర ఉప ముఖ్యమంత్రి శ్రీ జిష్ణు దేవ్ వర్మ గారు, నా క్యాబినెట్ సహచరులు సోదరి ప్రతిమా భౌమిక్ జీ మరియు శ్రీ జ్యోతిరాదిత్య సింధియా జీ, రాష్ట్ర ప్రభుత్వ మంత్రులు శ్రీ ఎన్ సి డెబర్మా జీ, శ్రీ రతన్ లాల్ నాథ్ జీ, శ్రీ ప్రాంజిత్ సింఘా రాయ్ జీ మరియు శ్రీ మనోజ్ కాంతి దేబ్ జీ, ఇతర ప్రజ ప్రతినిధులు, పెద్ద సంఖ్యలో వచ్చిన నా ప్రియమైన సోదర సోదరీమణులు! మీ అందరికీ శుభాకాంక్షలు. 2022 కొత్త సంవత్సరం సందర్భంగా మీ అందరికీ నా శుభాకాంక్షలు! సంవత్సరం ప్రారంభంలోనే, మా త్రిపుర సుందరి ఆశీస్సులతో త్రిపుర ఈ రోజు మూడు కానుకలు అందుకుంటుంది. ...

**te headline:** త్రిపురలో మహారాజ్ బీర్ బిక్రమ్ విమానాశ్రయం, ఇతర ప్రాజెక్టుల ప్రారంభోత్సవం సందర్భంగా ప్రధానమంత్రి ప్రసంగం పాఠం

---

Table 17: Two examples of PMIndiaSum document-headline pairs in English, Hindi, and Telugu. The documents' first sentences could be part of a speech, which poses a problem when used as a summary, whereas the headline is a more appropriate summary.

| Error type | Error description | | |
|---|---|---|---|
| Comprehensibility | **The output cannot be understood: wrong language, empty string, etc.** | | |
| | E.g. | Reference: | Cabinet approves creation of one post each of Vice-Chairperson and Member in the National |
| | | Output: | Cabinet approves creation of two-on-one posts of the National Cleaning Employees Commission |
| Grammar & fluency | **The output has grammatical errors or is unnatural and awkward.** | | |
| | E.g. | Reference: | PM to dedicate naval submarine INS Kalvari to the nation tomorrow |
| | | Output: | PM to dedicate Navy Navy Navy Navy Navy Navy Navy Navy Navy Navy |
| Factuality | **The output contains incorrect details as per the reference: named entity, number, etc.** | | |
| | E.g. | Reference: | Cabinet apprised of two Bilateral MoUs between India and Cuba, and India and Korea in the area of Biotechnology |
| | | Output: | Cabinet apprised of the MoU between India and Mongolia on cooperation in the field of Biotechnology |
| Omission | **The output does not cover some information in the reference.** | | |
| | E.g. | Reference: | PM to hold 'Samvad' with Beneficiaries of Pradhan Mantri Bhartiya Janaushadhi Pariyojna and affordable cardiac stents and knee implants on June 7 |
| | | Output: | PM to interact with beneficiaries of Pradhan Mantri Bhartiya Janaushadhi Pariyojna on 7th June |
| Redundancy | **The output consists of repeated or too much information compared to the reference.** | | |
| | E.g. | Reference: | PM to confer Awards for Excellence in Public Administration and address Civil Servants tomorrow |
| | | Output: | PM to confer Awards for Excellence in Public Administration for effective implementation of identified Priority Programs and Innovation to districts/implementing units and other Central/State organisations at Vigyan Bhawan tomorrow |
| No error | **The output conveys the same information as the reference.** | | |

Table 18: Error categories defined in our error analysis.

**Table 19: Cross-lingual benchmarks: standard deviations.**

| | Summarization-Translation | | | | | | Translation-Summarization | | | | | | Fine-tuning | | | | | | Zero-shot | | | | | |
|---|---|---|---|---|---|---|---|---|---|---|---|---|---|---|---|---|---|---|---|---|---|---|---|---|
| | IndicBART | | | mBART | | | IndicBART | | | mBART | | | IndicBART | | | mBART | | | IndicBART | | | mBART | | |
| | R2 | RL | BL | R2 | RL | BL | R2 | RL | BL | R2 | RL | BL | R2 | RL | BL | R2 | RL | BL | R2 | RL | BL | R2 | RL | BL |
| hi-en | 0.4 | 0.5 | 0.6 | 0.5 | 0.4 | 0.6 | 0.5 | 0.6 | 0.7 | 0.9 | 1.1 | 1.2 | 0.3 | 0.2 | 0.7 | 6.9 | 6.0 | 5.9 | 0.0 | 0.1 | 0.0 | 0.4 | 4.3 | 0.1 |
| en-hi | 0.4 | 0.8 | 0.5 | 0.3 | 0.5 | 0.5 | 0.8 | 0.8 | 0.9 | 1.3 | 1.0 | 1.2 | 0.3 | 0.2 | 0.4 | 1.8 | 1.8 | 1.9 | 0.0 | 0.1 | 0.0 | 0.0 | 0.2 | 0.0 |
| gu-te | 0.2 | 0.4 | 0.0 | 0.3 | 0.4 | 0.2 | 0.4 | 0.4 | 0.4 | 0.7 | 1.1 | 0.3 | 0.4 | 0.5 | 0.2 | 0.5 | 1.1 | 0.2 | 0.0 | 0.0 | 0.0 | 0.0 | 0.1 | 0.0 |
| te-gu | 0.5 | 0.2 | 0.3 | 0.6 | 1.0 | 0.2 | 0.1 | 0.4 | 0.3 | 0.9 | 1.9 | 0.5 | 0.3 | 0.2 | 0.3 | 0.7 | 0.2 | 0.2 | 0.0 | 0.0 | 0.0 | 0.0 | 0.1 | 0.0 |
| ml-mni | 0.4 | 0.5 | 0.4 | 1.5 | 1.6 | 1.3 | 0.4 | 0.3 | 0.5 | 0.4 | 0.9 | 0.5 | 0.2 | 0.4 | 0.3 | 1.1 | 1.5 | 0.3 | 0.0 | 0.0 | 0.0 | 0.0 | 0.0 | 0.0 |
| mni-ml | 0.7 | 0.6 | 0.6 | 1.5 | 2.0 | 0.9 | 0.5 | 0.6 | 0.3 | 0.1 | 0.2 | 0.1 | 0.3 | 0.3 | 0.1 | 0.8 | 3.1 | 0.3 | 0.0 | 0.0 | 0.0 | 0.0 | 0.0 | 0.0 |
| mr-bn | 0.7 | 0.4 | 0.4 | 1.5 | 1.5 | 1.1 | 0.3 | 0.1 | 0.1 | 0.4 | 0.0 | 0.7 | 0.2 | 0.2 | 0.0 | 1.1 | 1.9 | 1.2 | 0.0 | 0.0 | 0.0 | 0.0 | 0.0 | 0.0 |
| bn-mr | 0.2 | 0.2 | 0.3 | 0.8 | 1.3 | 0.4 | 0.7 | 0.8 | 0.5 | 1.1 | 1.8 | 0.6 | 0.2 | 0.1 | 0.0 | 1.7 | 2.4 | 2.3 | 0.0 | 0.0 | 0.0 | 0.0 | 0.0 | 0.0 |
| te-ta | 0.2 | 0.2 | 0.1 | 0.7 | 0.8 | 0.5 | 0.2 | 0.7 | 0.5 | 1.1 | 1.8 | 0.6 | 0.2 | 0.1 | 0.0 | 1.7 | 2.4 | 2.3 | 0.0 | 0.0 | 0.0 | 0.0 | 0.0 | 0.0 |
| ta-te | 0.5 | 0.6 | 0.5 | 0.6 | 0.4 | 0.4 | 0.9 | 1.0 | 0.4 | 1.2 | 1.1 | 0.3 | 0.4 | 0.6 | 0.4 | 1.3 | 2.2 | 3.3 | 0.0 | 0.0 | 0.0 | 0.4 | 2.7 | 0.0 |
| mni-en | 0.5 | 0.6 | 0.3 | 0.2 | 0.6 | 0.3 | 0.1 | 0.1 | 0.1 | 1.5 | 1.2 | 0.9 | 0.5 | 0.5 | 0.4 | 6.0 | 9.4 | 4.3 | 0.0 | 0.0 | 0.0 | 0.0 | 0.0 | 0.0 |
| en-mni | 0.6 | 0.8 | 0.6 | 0.4 | 0.6 | 0.7 | 0.3 | 0.2 | 0.2 | 1.7 | 2.8 | 1.0 | 0.2 | 0.3 | 0.6 | 1.4 | 3.5 | 0.6 | 0.0 | 0.0 | 0.0 | 0.0 | 0.0 | 0.0 |

**Table 20: Multilingual benchmark with IndicBART: standard deviations.**

| | as | | | bn | | | gu | | | hi | | | kn | | | ml | | | mni | | | mr | | | or | | | pa | | | ta | | | te | | | en | | |
|---|---|---|---|---|---|---|---|---|---|---|---|---|---|---|---|---|---|---|---|---|---|---|---|---|---|---|---|---|---|---|---|---|---|---|---|---|---|---|---|
| | R2 | RL | BL | R2 | RL | BL | R2 | RL | BL | R2 | RL | BL | R2 | RL | BL | R2 | RL | BL | R2 | RL | BL | R2 | RL | BL | R2 | RL | BL | R2 | RL | BL | R2 | RL | BL | R2 | RL | BL | R2 | RL | BL |
| as | 0.8 | 0.9 | 0.4 | 0.5 | 0.5 | 0.4 | 1.3 | 2.1 | 0.8 | 0.3 | 0.3 | 0.1 | 0.3 | 0.5 | 0.2 | 0.2 | 0.3 | 0.3 | 0.7 | 1.6 | 1.2 | 0.5 | 0.7 | 0.2 | 0.7 | 1.4 | 0.3 | 0.3 | 0.7 | 0.4 | 0.5 | 0.8 | 0.4 | 0.7 | 1.1 | 0.2 |
| bn | 0.6 | 1.2 | 0.3 | 0.6 | 0.9 | 0.3 | 0.7 | 1.7 | 0.4 | 0.8 | 0.8 | 0.6 | 0.6 | 1.5 | 0.6 | 0.5 | 1.1 | 1.1 | 0.4 | 0.2 | 0.2 | 0.6 | 0.6 | 0.6 | 0.7 | 0.7 | 0.3 | 1.2 | 1.7 | 0.8 | 1.8 | 1.9 | 0.7 | 0.4 | 0.8 | 0.3 |
| gu | 0.1 | 0.1 | 0.1 | 0.3 | 0.5 | 0.1 | 0.4 | 0.7 | 0.5 | 1.6 | 1.3 | 1.2 | 0.7 | 0.3 | 0.3 | 0.7 | 0.1 | 0.3 | 0.6 | 1.2 | 0.5 | 0.9 | 1.3 | 0.3 | 0.9 | 1.3 | 0.3 | 0.3 | 0.4 | 0.3 | 1.9 | 1.7 | 1.2 | 1.2 | 2.1 | 0.3 |
| hi | 0.6 | 1.1 | 0.4 | 0.2 | 0.6 | 0.0 | 0.4 | 0.9 | 0.3 | 0.7 | 0.4 | 0.7 | 1.7 | 2.3 | 1.1 | 0.8 | 0.4 | 0.8 | 0.4 | 0.4 | 0.4 | 1.7 | 2.5 | 1.1 | 1.4 | 2.3 | 1.1 | 0.6 | 0.9 | 0.6 | 1.6 | 1.6 | 0.5 | 0.7 | 0.6 | 0.3 |
| kn | 0.3 | 0.5 | 0.2 | 0.3 | 0.3 | 0.0 | 0.9 | 0.4 | 0.6 | 0.9 | 1.7 | 1.1 | 0.8 | 1.0 | 0.8 | 1.0 | 0.7 | 1.2 | 0.1 | 0.8 | 0.6 | 0.4 | 0.9 | 0.8 | 1.0 | 1.5 | 0.8 | 1.0 | 1.0 | 0.3 | 0.8 | 1.8 | 0.8 | 1.4 | 2.5 | 0.6 |
| ml | 0.2 | 0.7 | 0.1 | 0.7 | 1.2 | 0.2 | 0.7 | 1.5 | 0.4 | 0.5 | 0.8 | 0.5 | 1.3 | 0.6 | 0.5 | 0.5 | 0.8 | 1.2 | 0.6 | 0.5 | 0.2 | 1.1 | 1.9 | 1.4 | 0.4 | 1.4 | 1.4 | 0.3 | 0.6 | 0.1 | 1.0 | 0.8 | 0.8 | 0.4 | 0.2 | 0.3 |
| mni | 0.1 | 0.1 | 0.0 | 0.0 | 0.1 | 0.0 | 0.2 | 0.3 | 0.1 | 0.5 | 0.8 | 0.5 | 0.5 | 0.6 | 0.6 | 0.0 | 0.0 | 0.0 | 0.2 | 0.6 | 0.2 | 1.2 | 1.5 | 0.0 | 0.1 | 0.3 | 0.0 | 0.1 | 0.4 | 0.3 | 1.0 | 0.4 | 0.3 | 0.8 | 1.0 | 0.3 |
| mr | 0.2 | 0.4 | 0.1 | 0.6 | 1.4 | 0.3 | 0.6 | 1.1 | 0.3 | 0.4 | 0.3 | 0.2 | 1.9 | 2.1 | 1.2 | 0.6 | 0.9 | 2.1 | 0.6 | 0.2 | 0.6 | 1.0 | 1.3 | 0.3 | 1.5 | 2.1 | 0.7 | 0.6 | 0.4 | 0.0 | 0.8 | 0.7 | 0.4 | 1.6 | 3.2 | 0.1 |
| or | 0.2 | 0.4 | 0.2 | 0.6 | 0.6 | 0.1 | 0.6 | 1.2 | 0.6 | 1.4 | 1.1 | 1.2 | 0.7 | 1.2 | 0.7 | 0.2 | 0.2 | 0.2 | 1.0 | 1.3 | 1.0 | 0.9 | 0.7 | 0.2 | 0.8 | 0.6 | 0.7 | 0.5 | 0.4 | 0.4 | 0.3 | 0.7 | 0.3 | 0.2 | 0.5 | 0.3 |
| pa | 0.3 | 0.4 | 0.3 | 0.2 | 0.6 | 0.0 | 0.9 | 1.3 | 0.3 | 1.6 | 1.4 | 1.6 | 0.8 | 0.9 | 0.9 | 0.1 | 0.4 | 0.7 | 0.8 | 0.8 | 0.5 | 0.6 | 0.8 | 0.6 | 0.5 | 1.0 | 0.2 | 0.2 | 0.5 | 0.3 | 1.8 | 1.8 | 1.0 | 2.7 | 3.8 | 0.1 |
| ta | 0.2 | 0.5 | 0.1 | 0.2 | 0.2 | 0.0 | 1.1 | 1.4 | 0.8 | 3.1 | 4.3 | 2.1 | 0.2 | 0.2 | 0.2 | 0.1 | 0.1 | 0.4 | 0.9 | 0.8 | 0.6 | 0.4 | 0.8 | 0.3 | 0.6 | 0.8 | 0.3 | 1.0 | 1.6 | 1.3 | 0.8 | 0.6 | 0.2 | 1.0 | 1.4 | 0.1 |
| te | 0.1 | 0.2 | 0.0 | 0.1 | 0.1 | 0.0 | 0.1 | 0.5 | 0.1 | 1.6 | 2.2 | 0.6 | 0.7 | 0.4 | 0.7 | 0.9 | 0.2 | 0.7 | 0.2 | 0.5 | 0.0 | 0.2 | 0.5 | 0.3 | 0.8 | 0.8 | 0.3 | 0.6 | 0.8 | 0.2 | 1.1 | 0.7 | 0.6 | 1.0 | 1.6 | 0.6 |
| en | 0.2 | 0.5 | 0.1 | 0.1 | 0.9 | 0.0 | 0.8 | 1.1 | 0.3 | 0.3 | 0.5 | 0.2 | 0.2 | 0.9 | 0.2 | 0.5 | 0.5 | 0.6 | 0.1 | 0.9 | 0.3 | 0.7 | 1.1 | 0.9 | 0.4 | 1.1 | 0.9 | 1.0 | 2.1 | 0.3 | 1.1 | 1.1 | 0.3 | 1.1 | 1.2 | 1.3 |

**Table 21: Multilingual benchmark with mBART: standard deviations.**

| | bn | | | gu | | | hi | | | ml | | | mni | | | mr | | | ta | | | te | | | ur | | | en | | |
|---|---|---|---|---|---|---|---|---|---|---|---|---|---|---|---|---|---|---|---|---|---|---|---|---|---|---|---|---|---|---|
| | R2 | RL | BL | R2 | RL | BL | R2 | RL | BL | R2 | RL | BL | R2 | RL | BL | R2 | RL | BL | R2 | RL | BL | R2 | RL | BL | R2 | RL | BL | R2 | RL | BL |
| bn | 0.6 | 0.9 | 0.6 | 0.3 | 0.2 | 0.5 | 0.5 | 0.5 | 0.2 | 0.8 | 1.8 | 0.5 | 1.1 | 0.6 | 1.0 | 1.0 | 0.5 | 0.9 | 1.1 | 0.9 | 1.0 | 0.8 | 0.4 | 0.5 | 1.4 | 0.9 | 1.5 | 2.1 | 1.6 | 1.2 |
| gu | 2.0 | 1.8 | 1.2 | 0.4 | 0.8 | 1.4 | 0.6 | 0.4 | 1.4 | 2.3 | 2.5 | 1.1 | 0.3 | 0.7 | 0.2 | 1.6 | 0.7 | 1.3 | 1.1 | 0.5 | 1.4 | 1.9 | 1.7 | 1.8 | 0.6 | 1.1 | 0.9 | 1.7 | 1.1 | 0.8 |
| hi | 1.3 | 1.5 | 0.6 | 0.9 | 1.2 | 0.5 | 1.6 | 1.6 | 0.5 | 1.7 | 1.7 | 0.8 | 0.6 | 0.3 | 0.3 | 0.8 | 0.8 | 0.8 | 0.8 | 0.4 | 1.1 | 1.9 | 1.2 | 1.9 | 1.0 | 0.8 | 1.2 | 2.4 | 1.3 | 2.3 |
| ml | 1.1 | 0.8 | 0.5 | 1.3 | 1.3 | 0.9 | 0.1 | 0.3 | 0.5 | 1.8 | 1.6 | 1.2 | 2.0 | 2.0 | 1.3 | 2.4 | 1.9 | 1.7 | 2.7 | 2.0 | 2.4 | 1.6 | 1.8 | 1.6 | 1.0 | 1.0 | 0.7 | 3.6 | 3.0 | 2.4 |
| mni | 0.5 | 0.6 | 0.4 | 0.3 | 0.5 | 0.4 | 0.6 | 0.7 | 0.6 | 2.1 | 2.3 | 1.5 | 0.4 | 0.4 | 0.3 | 1.5 | 1.4 | 1.5 | 1.3 | 1.1 | 1.2 | 0.8 | 0.8 | 0.8 | 1.0 | 1.1 | 1.0 | 2.9 | 2.2 | 2.6 |
| mr | 1.5 | 1.5 | 1.0 | 1.2 | 1.5 | 1.0 | 1.0 | 0.8 | 1.0 | 1.8 | 2.2 | 0.9 | 0.6 | 0.6 | 0.7 | 1.8 | 1.1 | 1.9 | 1.4 | 1.4 | 1.4 | 1.0 | 0.4 | 1.0 | 0.7 | 0.3 | 0.5 | 0.9 | 0.6 | 1.1 |
| ta | 1.6 | 1.5 | 0.7 | 1.5 | 1.1 | 0.7 | 0.8 | 1.1 | 0.7 | 1.3 | 1.5 | 0.6 | 1.0 | 0.6 | 1.0 | 0.5 | 0.5 | 0.4 | 1.0 | 1.0 | 1.0 | 0.3 | 0.4 | 0.3 | 1.1 | 1.1 | 1.0 | 2.1 | 1.7 | 0.7 |
| te | 1.2 | 1.3 | 0.1 | 0.7 | 0.2 | 1.0 | 0.2 | 0.1 | 1.0 | 0.9 | 0.6 | 1.4 | 0.6 | 0.9 | 0.6 | 1.7 | 1.1 | 1.7 | 1.8 | 1.5 | 1.8 | 0.6 | 0.7 | 0.6 | 0.3 | 0.8 | 0.3 | 2.1 | 1.2 | 0.4 |
| ur | 1.9 | 1.7 | 1.1 | 0.8 | 0.6 | 0.1 | 1.0 | 1.0 | 0.9 | 1.6 | 1.6 | 0.7 | 0.5 | 0.5 | 1.1 | 1.2 | 0.6 | 0.7 | 0.1 | 0.6 | 0.4 | 1.8 | 1.8 | 1.0 | 1.0 | 1.0 | 1.0 | 1.5 | 1.1 | 1.4 |
| en | 1.2 | 1.3 | 1.7 | 1.4 | 1.5 | 0.7 | 1.0 | 0.8 | 1.4 | 1.2 | 1.4 | 1.0 | 0.5 | 0.2 | 0.3 | 1.2 | 1.0 | 1.0 | 1.0 | 1.0 | 1.7 | 1.1 | 0.7 | 1.1 | 0.7 | 0.1 | 0.4 | 2.3 | 1.3 | 2.5 |

# I Datasheet for PMIndiaSum

## I.1 Motivation

**Q:** For what purpose was the dataset created? (Was there a specific task in mind? Was there a specific gap that needed to be filled? Please provide a description.)
**A:** The dataset was developed to facilitate research on monolingual, cross-lingual, and multilingual headline summarization, specifically for Indian languages. Given the nature of multilingualism in India, cross-lingual summarization can greatly ease information access, but there is currently little availability in such datasets that can be used for researching the topic. Refer to Section 1 and Section 5 for more details.

**Q:** Who created the dataset (e.g., which team, research group) and on behalf of which entity (e.g., company, institution, organization)?
**A:** Researchers from IIIT Hyderabad and the University of Edinburgh created the dataset.

**Q:** Who funded the creation of the dataset?
**A:** Please refer to the author list for the people who created this dataset; the funding sources for authors are stated in the acknowledgement section.

**Q:** Any other comments?
**A:** No.

## I.2 Composition

**Q:** What do the instances that comprise the dataset represent (e.g., documents, photos, people, countries)? (Are there multiple types of instances (e.g., movies, users, and ratings; people and interactions between them; nodes and edges)? Please provide a description.)
**A:** Each instance in the dataset contains a news article and corresponding headline either in the same or different languages. Refer to Table 17 for data samples.

**Q:** How many instances are there in total (of each type, if appropriate)?
**A:** The dataset consists of 76,680 monolingual and 620,336 cross-lingual document-headline pairs. Please refer to Table 13 for individual language pair counts.

**Q:** Does the dataset contain all possible instances or is it a sample (not necessarily random) of instances from a larger set?
**A:** The dataset consists of all instances derived from the raw data we gathered and processed.

**Q:** Is any information missing from individual instances? If so, please provide a description, explaining why this information is missing (e.g., because it was unavailable). This does not include intentionally removed information, but might include, e.g., redacted text.
**A:** No.

**Q:** Are relationships between individual instances made explicit (e.g., users' movie ratings, social network links)? If so, please describe how these relationships are made explicit.)
**A:** Yes. Data instances are mostly independent of each other. Data instances of document-headline pairs in different languages from the same article convey the same content in the original news articles.

**Q:** Are there recommended data splits (e.g., training, development/validation, testing)? If so, please provide a description of these splits, explaining the rationale behind them.
**A:** Yes. Refer to Section 2.5 for an explanation. The split information is in the data file itself. We also provide a code zip which has the script to create data splits linked in Footnote 1.

**Q:** Are there any errors, sources of noise, or redundancies in the dataset?
**A:** Our expectation is that the articles are professionally written with no errors, and we take measures to ensure that there are no redundancies by removing duplicated data. However, it is possible that extraneous HTML elements may introduce errors that remain unfiltered from the raw crawled data. It is not feasible to manually inspect all data instances or automatically identify this type of noise.

**Q:** Is the dataset self-contained, or does it link to or otherwise rely on external resources (e.g., websites, tweets, other datasets)? (If it links to or relies on external resources, a) are there guarantees that they will exist, and remain constant, over time; b) are there official archival versions of the complete dataset (i.e., including the external resources as they existed at the time the dataset was

created); c) are there any restrictions (e.g., licenses, fees) associated with any of the external resources that might apply to a dataset consumer? Please provide descriptions of all external resources and any restrictions associated with them, as well as links or other access points, as appropriate.)

**A:** The dataset is self-contained. The dataset can be downloaded, used, adapted, and re-distributed without restrictions.

**Q:** Does the dataset contain data that might be considered confidential (e.g., data that is protected by legal privilege or by doctor–patient confidentiality, data that includes the content of individuals' non-public communications)? If so, please provide a description.

**A:** No, as all articles in the dataset are publicly available from the PMIndia's website.

**Q:** Does the dataset contain data that, if viewed directly, might be offensive, insulting, threatening, or might otherwise cause anxiety? If so, please describe why.

**A:** Unlikely, but we cannot completely rule this out because the data contains news articles from a governmental website.

**Q:** Does the dataset relate to people? (If not, you may skip the remaining questions in this section.)

**A:** Yes, the majority of the articles are about the Prime Minister of India who is a public figure. The data also contains news about other real people.

**Q:** Does the dataset identify any subpopulations (e.g., by age, gender)? If so, please describe how these subpopulations are identified and provide a description of their respective distributions within the dataset.

**A:** While the dataset does not explicitly identify any subpopulations based on factors such as age or gender, it is possible that such information may be mentioned in the news articles themselves, such as when referring to specific individuals. As news articles are likely to include details such as gender, age, occupation, etc., it is possible that these factors may be indirectly associated with the data instances in the dataset. However, we do not have any explicit information on the distributions of these subpopulations within the dataset.

**Q:** Is it possible to identify individuals (i.e., one or more natural persons), either directly or indirectly (i.e., in combination with other data) from the dataset? If so, please describe how.

**A:** Yes, there are individuals' names present in the data.

**Q:** Does the dataset contain data that might be considered sensitive in any way (e.g., data that reveals race or ethnic origins, sexual orientations, religious beliefs, political opinions or union memberships, or locations; financial or health data; biometric or genetic data; forms of government identification, such as social security numbers; criminal history)? If so, please provide a description.

**A:** While it is unlikely that the dataset contains sensitive information that is not already public, there is a possibility that certain news articles in the dataset may contain details that could be considered sensitive. For example, news articles may refer to individuals' personal information. However, as our dataset is derived from a public governmental news website, any sensitive information that may be present in the dataset is likely to have already been publicly disclosed.

**Q:** Any other comments?
**A:** No.

### I.3 Collection process

**Q:** How was the data associated with each instance acquired? (Was the data directly observable (e.g., raw text, movie ratings), reported by subjects (e.g., survey responses), or indirectly inferred/derived from other data (e.g., part-of-speech tags, model-based guesses for age or language)? If data was reported by subjects or indirectly inferred/derived from other data, was the data validated/verified? If so, please describe how.)

**A:** The data is crawled from the Prime Minister of India website followed by processing. It is observable on the website. The data is reported directly by the website.

**Q:** What mechanisms or procedures were used to collect the data (e.g., hardware apparatus or sensor, manual human curation, software program, software API)? (How were these mechanisms or procedures validated?)

**A:** The data was scraped from a website using a

specifically-design open-source crawler and parser. Please refer to Section 2.1 for more details. We have manually inspected a few samples to verify the sample quality and parallelism.

**Q:** If the dataset is a sample from a larger set, what was the sampling strategy (e.g., deterministic, probabilistic with specific sampling probabilities)?
**A:** The dataset is not sampled from a larger corpus.

**Q:** Who was involved in the data collection process (e.g., students, crowd workers, contractors) and how were they compensated (e.g., how much were crowd workers paid)?
**A:** The data set is crawled and processed automatically. The scripts to crawl and process data are partly open-source, and partly written by the authors of this paper.

**Q:** Over what timeframe was the data collected? (Does this timeframe match the creation timeframe of the data associated with the instances (e.g., recent crawl of old news articles)? If not, please describe the timeframe in which the data associated with the instances was created.)
**A:** The data was crawled in early 2023. The crawled data span articles published between 2014 and early 2023.

**Q:** Were any ethical review processes conducted (e.g., by an institutional review board)? If so, please provide a description of these review processes, including the outcomes, as well as a link or other access point to any supporting documentation.
**A:** No.

**Q:** Did you collect the data from the individuals in question directly, or obtain it via third parties or other sources (e.g., websites)?
**A:** The dataset was obtained from a third-party website that publishes data or information of individuals.

**Q:** Were the individuals in question notified about the data collection? (If so, please describe (or show with screenshots or other information) how notice was provided, and provide a link or other access point to, or otherwise reproduce, the exact language of the notification itself.)

**A:** No. The news articles are public.

**Q:** Did the individuals in question consent to the collection and use of their data? (If so, please describe (or show with screenshots or other information) how consent was requested and provided, and provide a link or other access point to, or otherwise reproduce, the exact language to which the individuals consented.)
**A:** No. The news articles are public.

**Q:** If consent was obtained, were the consenting individuals provided with a mechanism to revoke their consent in the future or for certain uses? (If so, please provide a description, as well as a link or other access point to the mechanism (if appropriate).)
**A:** N/A.

**Q:** Has an analysis of the potential impact of the dataset and its use on data subjects (e.g., a data protection impact analysis) been conducted? (If so, please provide a description of this analysis, including the outcomes, as well as a link or other access point to any supporting documentation.)
**A:** No.

**Q:** Any other comments?
**A:** No.

### I.4 Preprocessing, cleaning, labeling

**Q:** Was any preprocessing/cleaning/labeling of the data done (e.g., discretization or bucketing, tokenization, part-of-speech tagging, SIFT feature extraction, removal of instances, processing of missing values)? (If so, please provide a description. If not, you may skip the remainder of the questions in this section.)
**A:** Yes, detailed in Section 2.2.

**Q:** Was the "raw" data saved in addition to the preprocessed/cleaned/labeled data (e.g., to support unanticipated future uses)? (If so, please provide a link or other access point to the "raw" data.)
**A:** The "raw" data is saved but not made public at the moment. We plan to do so shortly.

**Q:** Is the software used to preprocess/clean/label the instances available? (If so, please provide a link or other access point.)

**A:** We uploaded a code zip which has these scripts.

**Q:** Any other comments?
**A:** No.

### I.5 Uses

**Q:** Has the dataset been used for any tasks already? (If so, please provide a description.)
**A:** The dataset has been used by ourselves to fine-tune existing models to perform the headline summarization task. See Section 3 for more details.

**Q:** Is there a repository that links to any or all papers or systems that use the dataset? (If so, please provide a link or other access point.)
**A:** No.

**Q:** What (other) tasks could the dataset be used for?
**A:** The dataset can be utilized for a wide range of NLP tasks concerning languages in India: language modelling, named entity recognition, linguistic analysis, etc.

**Q:** Is there anything about the composition of the dataset or the way it was collected and preprocessed/cleaned/labeled that might impact future uses? (For example, is there anything that a future user might need to know to avoid uses that could result in unfair treatment of individuals or groups (e.g., stereotyping, quality of service issues) or other undesirable harms (e.g., financial harms, legal risks) If so, please provide a description. Is there anything a future user could do to mitigate these undesirable harms?)
**A:** Yes, we applied strict language filtering which removed code-mixing instances, however, it could be a common and valid language phenomenon in languages of India.

**Q:** Are there tasks for which the dataset should not be used? (If so, please provide a description.)
**A:** Our dataset, like any large dataset, has the potential to be used in harmful ways. One example of such harm is the creation of models that generate hate speech or fake news using the data. Additionally, as previously mentioned, the dataset may contain sensitive information that should not be used for discriminatory or harmful purposes. It is important to carefully consider the intended use

of the dataset and ensure that it aligns with ethical and legal standards.

**Q:** Any other comments?
**A:** No.

### I.6 Distribution

**Q:** Will the dataset be distributed to third parties outside of the entity (e.g., company, institution, organization) on behalf of which the dataset was created? (If so, please provide a description.)
**A:** Yes, the data will be free to the public to download, use, modify and re-distribute.

**Q:** How will the dataset be distributed (e.g., tarball on website, API, GitHub)? (Does the dataset have a digital object identifier (DOI)?)
**A:** The dataset is currently hosted on Hugging Face as indicated in Footnote 1, as well as on Google Drive as a zip file.

**Q:** When will the dataset be distributed?
**A:** The dataset is available now.

**Q:** Will the dataset be distributed under a copyright or other intellectual property (IP) license, and/or under applicable terms of use (ToU)? (If so, please describe this license and/or ToU, and provide a link or other access point to, or otherwise reproduce, any relevant licensing terms or ToU, as well as any fees associated with these restrictions.)
**A:** Yes, the dataset is distributed under the CC BY 4.0 license.

**Q:** Have any third parties imposed IP-based or other restrictions on the data associated with the instances? (If so, please describe these restrictions, and provide a link or other access point to, or otherwise reproduce, any relevant licensing terms, as well as any fees associated with these restrictions.).
**A:** The original website states that "the material has to be reproduced accurately and not to be used in a derogatory manner or in a misleading context. Wherever the material is being published or issued to others, the source must be prominently acknowledged." More information is available at https://www.pmindia.gov.in/en/website-policies/

**Q:** Do any export controls or other regulatory restrictions apply to the dataset or individual

instances? (If so, please describe these restrictions, and provide a link or other access point to, or otherwise reproduce, any supporting documentation.)

**A:** Not known.

**Q:** Any other comments?
**A:** No.

### I.7 Maintenance

**Q:** Who is supporting/hosting/maintaining the dataset?
**A:** Authors of this paper.

**Q:** How can the owner/curator/manager of the dataset be contacted (e.g., email address)?
**A:** Via email or issues in the Hugging Face or GitHub repositories as Footnote 1.

**Q:** Is there an erratum? (If so, please provide a link or other access point.)
**A:** No.

**Q:** Will the dataset be updated (e.g., to correct labeling errors, add new instances, delete instances)? (If so, please describe how often, by whom, and how updates will be communicated to users (e.g., mailing list, GitHub)?)
**A:** Currently there is no plan to update the dataset.

**Q:** If the dataset relates to people, are there applicable limits on the retention of the data associated with the instances (e.g., were individuals in question told that their data would be retained for a fixed period of time and then deleted)? (If so, please describe these limits and explain how they will be enforced.)
**A:** No.

**Q:** Will older versions of the dataset continue to be supported/hosted/maintained? (If so, please describe how. If not, please describe how its obsolescence will be communicated to users.)
**A:** There is no older version of the dataset. Its obsolescence will be communicated to users at the venue where it is held.

**Q:** If others want to extend/augment/build on/contribute to the dataset, is there a mechanism for them to do so? (If so, please provide a description. Will these contributions be validated/verified?

If so, please describe how. If not, why not? Is there a process for communicating/distributing these contributions to other users? If so, please provide a description.)
**A:** Yes, they can freely do so by any means as long as they abide by the license.

**Q:** Any other comments?
**A:** No.