# OpenReview forum: "PMIndiaSum: Multilingual and Cross-lingual Headline Summarization for Languages in India"
_EMNLP/2023/Conference — EMNLP 2023 Findings_

### Official Review · Reviewer_Jf8L · 2023-07-31

**Soundness:** 4

**Excitement:**

4: Strong: This paper deepens the understanding of some phenomenon or lowers the barriers to an existing research direction.

**Paper Topic And Main Contributions:**

This paper presents PMIndiaSum, a multilingual and cross-lingual summarization dataset that supports 14 languages and 196 language pairs across 4 language families. In addition, the authors provide benchmark results for monolingual, cross-lingual, and multilingual summarization for different paradigms: fine-tuning, prompting, and translate-and-summarize.

**Questions For The Authors:**

Line 78-83: if I understand the data correctly, the default language of each article is English. This means all articles should at least be available in English. However, according to Table 2, English has 8160 articles, which is less than the sum of the number of articles in Figure 2. It would be great if the authors could clarify this to avoid misunderstanding.

Line 122-125: If the authors want to ensure the abstractive nature, why do authors only remove all samples where the summary is repeated as the initial or first few sentences in the document? There can be cases where the summary occurs at the middle or even the end of the document, right?

Table 3 shows that the first sentences can be considered as a summary for more than half of the articles out of 50 random samples. In Lines 122-125, the authors already removed quite a few clear repetitions. Does this suggest that many of the remained articles also have some kind of redundancy (headlines and the first sentences can both be considered as summaries)?

Line 338-340: as no PLM support mni language, does the author also train a tokenizer and subword embeddings on monolingual data for mni and merge with the original model?

Line 342-346: The author uses a public external translator to perform the translation in the summarization-and-translation workflow. I have doubts here. Then the cross-lingual performance would be highly dependent on the quality of the external translator. How does the author ensure that, given many languages are indeed low-resource ones?

Table 4: Oracle performance on **te** language is significantly higher than the other models, does the author have any explanation for that?


**Reasons To Accept:**

- This paper is well-written, well-documented, and easy to follow.

- This paper contributes to the summarization by presenting a new dataset that focuses on Indian languages (including low-resource ones).

- This author conducts extensive experiments and provides several benchmark results under different settings.


**Reasons To Reject:**

- The dataset is domain-specific and biased toward political news. Therefore it might be questionable whether it could be used as a proper dataset if one wants to evaluate the general capability of a PLM.

- Although the author conducts many experiments, some analyses are not detailed enough (I guess it is due to the page limit). I would suggest the author go into more detail about the analysis in the camera-ready version.


**Reproducibility:**

4: Could mostly reproduce the results, but there may be some variation because of sample variance or minor variations in their interpretation of the protocol or method.

**Reviewer Confidence:**

3: Pretty sure, but there's a chance I missed something. Although I have a good feel for this area in general, I did not carefully check the paper's details, e.g., the math, experimental design, or novelty.

---

> ### Author Rebuttal · Authors · 2023-08-26
>
> **1. Reason to reject: The dataset is domain-specific and biased toward political news. Therefore it might be questionable whether it could be used as a proper dataset if one wants to evaluate the general capability of a PLM.**
>
> Author response:
>
> - We would like to clarify that the goal of our work is not to evaluate the capability of a PLM, but to enable research on multilingual and cross-lingual summarization for low-resource Indian languages (pairs). We used the two PLMs because we think they provide a good model initialization. On the other hand, an ideal general domain PLM should handle all domains including ours.
>
> **2. Reason to reject: Although the author conducts many experiments, some analyses are not detailed enough (I guess it is due to the page limit). I would suggest the author go into more detail about the analysis in the camera-ready version.**
>
> Author response:
>
> - Indeed we are struggling a bit with the page limit. Since this is a dataset contribution, we tried to split the analysis work on both the dataset **itself** (statistics, quality assurance, human inspection), as well as on experimental results (multi/cross/mono-lingual conditions, different PLMs, model error analysis).
>
> - We plan to expand our discussions on the PLM data conditions. It would be interesting to add analysis on domains, such as in-domain and out-of-domain data experiments, inspired by your comment on domains.
>
> **3. Question: Line 78-83: if I understand the data correctly, the default language of each article is English. This means all articles should at least be available in English. However, according to Table 2, English has 8160 articles, which is less than the sum of the number of articles in Figure 2. It would be great if the authors could clarify this to avoid misunderstanding.**
>
> Author response:
> - When constructing the dataset, we carried out a manual inspection and found that articles labeled as English could contain non-English content. Although there is an English URL for each article, **the default article language may not necessarily be English**. Essentially the URLs served as pivots/references for us to match cross-lingual data pairs, but we did not assume the content language. Therefore, we have strictly cleaned our data by language (lines 107-112 and Table 9 in Appendix B).
>
> **4. Questions: Line 122-125: If the authors want to ensure the abstractive nature, why do authors only remove all samples where the summary is repeated as the initial or first few sentences in the document? There can be cases where the summary occurs at the middle or even the end of the document, right?**
>
> Author response:
>
> - Initially, we followed previous works [1,2,3] to filter out data pairs where the summary is a prefix of the document. We hypothesize the reason behind this is that it removes the position bias where a model can simply overfit to copy the first sentence. For cases where the summary appears later, the model still needs to “understand” the document in order to “select” a summary. Here, such extraction becomes an easier (but valid) case of summarization.
>
> - Nonetheless, your suggestion is definitely a reasonable practice. We could filter out all data pairs where the summary matches a (or a few) sentences anywhere in the document, and update the data once the work can be de-anonymised.
>
> [1] MassiveSumm: a very large-scale, very multilingual, news summarisation dataset (Varab & Schluter, EMNLP 2021)
>
> [2] TeSum: Human-Generated Abstractive Summarization Corpus for Telugu (Urlana et al., LREC 2022)
>
> [3] LR-Sum: Summarization for Less-Resourced Languages (Palen-Michel & Lignos, Findings 2023)
>
> **5. Question: Table 3 shows that the first sentences can be considered as a summary for more than half of the articles out of 50 random samples. In Lines 122-125, the authors already removed quite a few clear repetitions. Does this suggest that many of the remained articles also have some kind of redundancy (headlines and the first sentences can both be considered as summaries)?**
>
> Author response:
>
> - Yes, it is totally possible that in the remaining data pairs, the first sentence in the document is a potential summary (but different from the headline summary). This will not hurt the downstream model because we have removed data pairs where the gold summary is the same as the prefix, so the model will not overfit by merely copying the first sentence.
>
> **6. Question: Line 338-340: as no PLM support mni language, does the author also train a tokenizer and subword embeddings on monolingual data for mni and merge with the original model?**
>
> Author response:
>
> - The Manipuri data is written in Bengali script as line 59 footnote 5. It is as released by the Indian government website and we did not make any transliteration. As such, as long as a PLM supports Bengali, it will be able to process our Manipuri data. We think your solution works better if the Manipuri data is written in Meitei script. We will improve the explanation in the experimental setup section.
>
> - As line 340, we randomly initialized a single embedding entry for the special token for mni because this is not included in either PLM’s vocabulary. This token acts as a language indicator and lets the model know the correct language to generate. The token is added to the vocabulary, and its embedding is merged with the PLMs' original embedding matrix.
>
> **7. Question: Line 342-346: The author uses a public external translator to perform the translation in the summarization-and-translation workflow. I have doubts here. Then the cross-lingual performance would be highly dependent on the quality of the external translator. How does the author ensure that, given many languages are indeed low-resource ones?**
>
> Author response:
>
> - Yes, the "translate-and-summarise" pipelines’ performance depends on the quality of the external translator. The translation engine we use is open-source and supports all the languages of interest. Yet it might be the quality bottleneck. **This is exactly the motivation behind our work**: the provision of cross-lingual data in 196 pairs removes the need for an external translator. This is evident in Table 6 where we compared fine-tuned models against "translate-and-summarise" approaches. We found that FT achieved similar scores without the need for an external translator.
>
> - Note that "translate-and-summarise" serves as a baseline, because without our work, cross-lingual summarization needs to rely on external translators.
>
> **8. Question: Table 4: Oracle performance on te language is significantly higher than the other models, does the author have any explanation for that?**
>
> Author response:
>
> - As seen in Table 4, the oracle scores for Telugu are in line with other languages’ oracle scores, thus it is not an issue with our dataset (e.g. no significant overlap between the reference summary and the document). On the other hand, the PLM fine-tuning scores look low for Telugu, but we think these are in the normal range. Referring to Table 6 in the [indicBART PLM paper](https://aclanthology.org/2022.findings-acl.145.pdf), Telugu summarization naturally has a lower score than other languages [4], perhaps due to its linguistic features, vocabulary, etc.
>
> [4] IndicBART: A Pre-trained Model for Indic Natural Language Generation (Dabre et al., Findings 2022)

---

### Official Review · Reviewer_2wEo · 2023-08-03

**Soundness:** 4

**Excitement:**

3: Ambivalent: It has merits (e.g., it reports state-of-the-art results, the idea is nice), but there are key weaknesses (e.g., it describes incremental work), and it can significantly benefit from another round of revision. However, I won't object to accepting it if my co-reviewers champion it.

**Paper Topic And Main Contributions:**

Paper presents the corpus construction of 14 languages focusing on Indian government articles. The 14 languages lead to 196 language pairs with a focus on summarization. The main contribution of the paper is the corpus itself which is by far the largest corpus for this purpose on languages used in India.

**Reasons To Accept:**

- Paper is well written and clear and easy to follow. The methodology is repeatable for researchers wanting to extend this work.
- The research follows standard experimentation and evaluation so it is comparable with work in the field
- The corpus itself is of sufficient size and scope to have an impact on this language pairs and the research community
- Supporting monolingual, cross-lingual, and multi-lingual summarization for under-resourced languages is a good contribution by the authors

**Reasons To Reject:**

- Some of the language pairs are covered by other corpora but I do not get a feel for how this would compare in the tasks with different data
- As noted by the authors the use of external translation models makes isolating the impact of this methodology and dataset difficult as that model may be and is likely ingesting additional data for some language pairs


**Reproducibility:**

5: Could easily reproduce the results.

**Reviewer Confidence:**

3: Pretty sure, but there's a chance I missed something. Although I have a good feel for this area in general, I did not carefully check the paper's details, e.g., the math, experimental design, or novelty.

---

> ### Author Rebuttal · Authors · 2023-08-26
>
> **1. Reason to reject: Some of the language pairs are covered by other corpora but I do not get a feel for how this would compare in the tasks with different data**
>
> Author response:
>
> - Some Indian languages are covered by other datasets, in summarization, mostly monolingual. Ours is the first massively parallel and cross-lingual corpus for 196 language pairs. Our motivation is not to improve model performance on other datasets, but more about facilitating research on non-English-centric multilingual and cross-lingual summarization research, especially for Indian languages with no existing resources before our work (e.g. imagine now one can train a model to directly summarise from Assamese to Manipuri).
>
> **2. Reason to reject: As noted by the authors the use of external translation models makes isolating the impact of this methodology and dataset difficult as that model may be and is likely ingesting additional data for some language pairs**
>
> Author response:
>
> - Admittedly comparison is not fair because a translation engine sees more data, but this is to the advantage of “translate-and-summarise”, which is a baseline without using our provided cross-lingual data.
>
> - We would like to clarify that your question is exactly the motivation for our work. Without PMIndiaSum, one can perform cross-lingual summarization **only** by adopting the “translate-and-summarise” approach, thus using more compute and data resources. Our work enables cross-lingual summarization without translation, achieving similar performance. In addition, the dataset could promote future research to better tackle the problem.
>
> - Further to your comment on "translate-and-summarise", we note that the translation process is a bottleneck itself, since erroneous translations lead to erroneous summaries. This is especially true for language pairs that do not have good translation engines in place due to a lack of parallel data, e.g. Assamese-Manipuri.

---

### Official Review · Reviewer_hbSQ · 2023-08-04

**Soundness:** 3

**Excitement:**

2: Mediocre: This paper makes marginal contributions (vs non-contemporaneous work), so I would rather not see it in the conference.

**Paper Topic And Main Contributions:**

This paper introduces PMIndiaSum, which is a cross-lingual headline summarization dataset covering 14 different Indian languages. The dataset was extracted from the Prime Minister of India website and contains over 76k monolingual document-headline pairs and 620k cross-lingual pairs in total.

The authors present a quantitative analysis of the dataset's qualities, including vocabulary size, compression ratio, summary novelty, and redundancy.

Finally the authors conduct fine-tuning of a summarization model over the dataset in both monolingual and cross-lingual settings and present the results.

The primary contribution here is the large multilingual corpus, and a set of statistics that may be used as the starting point for future research.

**Questions For The Authors:**

What is the utility of knowing compression of these document-summary pairs? Given the relatively limited headline size it only really gives us an indicator of the document length

**Reasons To Accept:**

- Large multilingual dataset for under-resourced language summarization
- Thorough set of descriptive statistics for the dataset

**Reasons To Reject:**

Unclear whether this is the appropriate venue for this paper; something like LREC might be a better fit.

**Reproducibility:**

4: Could mostly reproduce the results, but there may be some variation because of sample variance or minor variations in their interpretation of the protocol or method.

**Reviewer Confidence:**

3: Pretty sure, but there's a chance I missed something. Although I have a good feel for this area in general, I did not carefully check the paper's details, e.g., the math, experimental design, or novelty.

---

> ### Author Rebuttal · Authors · 2023-08-26
>
> **1. Reason to reject: Unclear whether this is the appropriate venue for this paper; something like LREC might be a better fit.**
>
> Author response:
> - We are encouraged to submit to this venue as the [EMNLP call for papers](https://2023.emnlp.org/calls/main_conference_papers/) welcomes "New data resources, particularly for low-resource languages". We also believe the two tracks "Resource and Evaluation" and  "Summarization" are appropriate for this work.
>
> - Our work provides a summarization dataset for 196 language pairs. Before this, the majority of the 196 pairs were in a “no-resource” state, despite having a large population of speakers. Without data resources, it would not be possible to test any empirical methods. We hope that you could kindly reconsider our contribution.
>
> **2. Question on compression**
>
> Author response:
>
> - Compression is a characteristic of a summarization dataset and is often reported by previous works [1,2,3]. Generally, it highlights the difficulty of content selection and aggregation imposed by the summarization task (the higher the more difficult). As in lines 151-153, the high compression of our dataset implies that the summaries are very abstractive and condensed.
>
> [1] Newsroom: A Dataset of 1.3 Million Summaries with Diverse Extractive Strategies (Grusky et al., NAACL 2018)
>
> [2] Intrinsic Evaluation of Summarization Datasets (Bommasani & Cardie, EMNLP 2020)
>
> [3] Models and Datasets for Cross-Lingual Summarisation (Perez-Beltrachini & Lapata, EMNLP 2021)

---

### Meta-Review · Area_Chair_VUXG · 2023-09-21

**Recommendation:** 4

**Metareview:**

In this study, the authors proposed multilingual headline summarization datasets. These datasets encompass 196 language pairs across 14 languages. The authors have made these datasets publicly available for the community. Reviewers mentioned that the dataset would be a valuable resource for the community and that the paper is well-written and easy to follow.

Reviewer jf8L expressed concerns about the need for a more detailed analysis of the experiments. These concerns should be addressed to give readers a comprehensive understanding of the experiments. Additional information can be included in the appendix.

---

### Decision · Program_Chairs · 2023-10-07

**Decision:**

Accept-Findings

**Comment:**

In this study, the authors proposed multilingual headline summarization datasets. These datasets encompass 196 language pairs across 14 languages. The authors have made these datasets publicly available for the community. Reviewers mentioned that the dataset would be a valuable resource for the community and that the paper is well-written and easy to follow.

Reviewer jf8L expressed concerns about the need for a more detailed analysis of the experiments. These concerns should be addressed to give readers a comprehensive understanding of the experiments. Additional information can be included in the appendix.